# GETS: Ensemble Temperature Scaling for Calibration in Graph Neural Networks

**Dingyi Zhuang**[*][†]
Massachusetts Institute of Technology
dingyi@mit.edu

**Chonghe Jiang**[†]
The Chinese University of Hong Kong
chjiang@link.cuhk.edu.hk

**Yunhan Zheng**
Singapore-MIT Alliance for Research and Technology (SMART)
yunhan.zheng@smart.mit.edu

**Shenhao Wang**
University of Florida
shenhaowang@ufl.edu

**Jinhua Zhao**
Massachusetts Institute of Technology
jinhua@mit.edu

## Abstract

Graph Neural Networks (GNNs) deliver strong classification results but often suffer from poor calibration performance, leading to overconfidence or underconfidence. This is particularly problematic in high-stakes applications where accurate uncertainty estimates are essential. Existing post-hoc methods, such as temperature scaling, fail to effectively utilize graph structures, while current GNN calibration methods often overlook the potential of leveraging diverse input information and model ensembles jointly. In this paper, we propose Graph Ensemble Temperature Scaling (GETS), a novel calibration framework that combines input and model ensemble strategies within a Graph Mixture-of-Experts (MoE) architecture. GETS integrates diverse inputs, including logits, node features, and degree embeddings, and adaptively selects the most relevant experts for each node's calibration procedure. Our method outperforms state-of-the-art calibration techniques, reducing expected calibration error (ECE) by $\geq 25\%$ across 10 GNN benchmark datasets. Additionally, GETS is computationally efficient, scalable, and capable of selecting effective input combinations for improved calibration performance. The implementation is available at https://github.com/ZhuangDingyi/GETS/.

## 1 Introduction

Graph Neural Networks (GNNs) have emerged as powerful tools for learning representations in numerous real-world applications, including social networks, recommendation systems, biological networks, and traffic systems, achieving state-of-the-art performance in tasks like node classification, link prediction, and graph classification (Kipf & Welling, 2016; Veličković et al., 2017). Their ability to capture complex relationships and dependencies makes them invaluable for modeling and predicting interconnected systems (Fan et al., 2019; Wu et al., 2019; Scarselli et al., 2008; Wu et al., 2022; 2021; Jiang* et al., 2024; Luo et al., 2024).

However, ensuring the reliability and trustworthiness of GNN prediction remains a critical challenge, especially in high-stakes applications where decisions can have significant consequences (e.g. human safety). Consider GNN classification as an example, one key aspect of refining model reliability is *uncertainty calibration*, which aims to align the predicted probabilities with the true likelihood of outcomes (Guo et al., 2017). In short, well-calibrated models provide confidence estimates that reflect real-world probabilities, which is essential for risk assessment and informed decision-making (Zhuang et al., 2023).

---

[*]Corresponding author.
[†]Equal contribution.

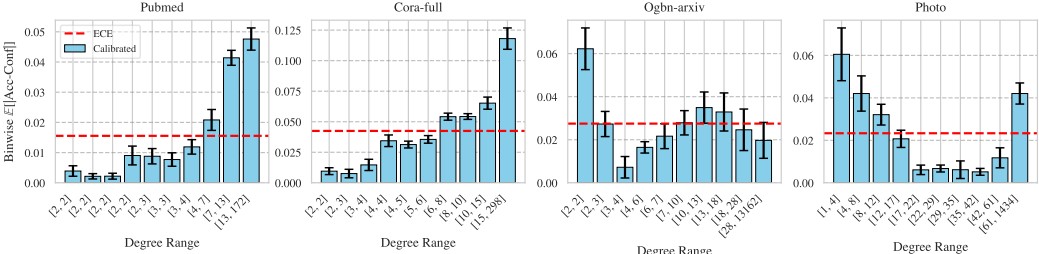

Figure 1: Expected calibration error (ECE), see Equation 3, for a well-trained CaGCN model (Wang et al., 2021). ECE is measured by grouping nodes based on degree ranges rather than predicted confidence. Datasets are sorted by average degree $\frac{2|\mathcal{E}|}{|\mathcal{V}|}$ as a measure of connectivity (see Table 1), from low to high. In lower-connectivity datasets like *Pubmed* and *Cora-full*, high-degree nodes tend to show larger calibration errors, whereas in more connected datasets like *Photo* and *Ogbn-arxiv*, low-degree nodes exhibit higher calibration errors.

Calibration methods are typically applied post-hoc and can be broadly classified into two categories: nonparametric and parametric. Nonparametric methods, such as histogram binning and isotonic regression, adjust confidence estimates based on observed data distributions without assuming a specific functional form (Zadrozny & Elkan, 2001; Naeini et al., 2015). While these methods allow flexibility in modeling, they often require large calibration datasets and can struggle with high-dimensional or unevenly distributed graph data, making them less practical for graphs with vast numbers of nodes. In contrast, parametric methods, including temperature scaling (TS) and Platt scaling, assume a functional relationship to adjust model outputs (Guo et al., 2017; Platt et al., 1999). Among these, TS has gained popularity due to its simplicity and effectiveness in calibrating models by smoothing predicted probabilities. This method is scalable and easy to implement, especially for large-scale graphs. Recent work has moved beyond the uniform adjustments applied to all data in classic TS, focusing on learning node-specific calibration parameters to capture individual characteristics that affect graph calibration performance (Tang et al., 2024; Hsu et al., 2022).

Motivated by the methodologies discussed above, when we dive into the GNN (classification) calibration task, we find that the subtleties lie in two main challenges: (i) the difficulty of representing the calibration error score for the individual sample points; (ii) the disability to identify and utilize multiple information related to the GNN calibration. Existing approaches for graph temperature scaling primarily rely on an additional GNN model to output node-wise temperatures. However, systematic methods for determining the inputs to the calibration network and for integrating various input information have not been thoroughly explored.

Given these challenges, we aim to address the following issues: (i) What are the key factors influencing the calibration task for GNN classification? (ii) How can we effectively integrate these factors, which may exist across different spaces and scales, to perform GNN calibration?

For issue (i), we observe that, in addition to the logits and features discussed in prior work (Wang et al., 2021; Huang et al., 2022), Figure 1 demonstrates that node degree influences the calibration error, consistent with the message-passing mechanism in GNNs. In some cases, a low node degree can result in worse calibration performance due to insufficient information propagation. Conversely, excessive (and potentially misleading) information may flood a single node, leading to overconfidence in its predictions (Wang et al., 2022). Regarding issue (ii), it remains unclear how to develop an effective algorithm that incorporates multiple factors for calibration within the graph structure. To address these challenges, we propose a novel calibration framework called **Graph Ensemble Temperature Scaling (GETS)**. Our approach integrates both input ensemble and model ensemble strategies within a Graph Mixture-of-Experts (MoE) architecture, thereby overcoming the limitations of existing methods. By incorporating multiple calibration experts—each specializing in different aspects of the influential factors affecting calibration—GETS can adaptively select and combine relevant information for each node.

Experimental results on 10 GNN benchmark datasets demonstrate that GETS consistently achieves superior calibration performance, significantly reducing the ECE across all datasets. This highlights its effectiveness in addressing the calibration problem in the GNN classification task. Additionally, GETS is scalable and capable of handling large-scale datasets with computational efficiency.

Our key contributions are summarized as:

- We introduce a novel framework, GETS, which combines input and model ensemble strategies within a Graph MoE architecture. This enables robust node-wise calibration by integrating diverse inputs, such as logits, features, and degree embeddings.

- GETS significantly reduces Expected Calibration Error (ECE) across 10 GNN benchmark datasets, outperforming state-of-the-art methods like CaGCN and GATS, and demonstrating effectiveness across diverse graph structures.

- GETS employs a sparse gating mechanism to select the top $k$ experts for each node, enabling scalable calibration on large graphs while maintaining computational efficiency.

## 2 RELATED WORK

### 2.1 UNCERTAINTY CALIBRATION

Uncertainty calibration adjusts the predicted probabilities of the model to more accurately reflect the true likelihood of outcomes. Calibration methods are generally classified into in-training and post-hoc approaches.

In-training calibration adjusts uncertainty during model training. Bayesian methods, such as Bayesian Neural Networks (BNNs) and variational inference, achieve this by placing distributions over model parameters (Gal & Ghahramani, 2016; Sensoy et al., 2018; Detommaso et al., 2022). Frequentist approaches, including conformal prediction (Zargarbashi et al., 2023; Huang et al., 2024; Deshpande et al., 2024) and quantile regression (Chung et al., 2021), are also employed to ensure well-calibrated outputs.

Despite these efforts, model misspecification and inaccurate inference often lead to miscalibration (Kuleshov et al., 2018). To address this, post-hoc methods recalibrate models after training. These can be divided into nonparametric and parametric approaches. Nonparametric methods, such as histogram binning (Zadrozny & Elkan, 2001), isotonic regression (Zadrozny & Elkan, 2002), and Dirichlet calibration (Kull et al., 2019), adjust predictions flexibly based on observed data (Naeini et al., 2014). Parametric methods assume a specific functional form and learn parameters that directly modify predictions. Examples include temperature scaling (Guo et al., 2017), Platt scaling (Platt et al., 1999), beta calibration (Kull et al., 2017), and so on.

### 2.2 GNN CALIBRATION

Calibration of GNNs remains an under-explored area (Zhuang et al., 2023). Take the node classification task as an example, most existing methods adopt post-hoc approaches, often relying on temperature scaling where the temperature is learned via a certain network.

Teixeira et al. (2019) noted that ignoring structural information leads to miscalibration in GNNs. GNNs also tend to be underconfident, unlike most multi-class classifiers, which are typically overconfident. To address this, CaGCN introduces a GCN layer on top of the base GNN for improved uncertainty estimation using graph structure (Wang et al., 2021). Hsu et al. (2022) identified key factors affecting GNN calibration and proposed GATS, which employs attention layers to account for these factors. Tang et al. (2024) further explored node similarity within classes and proposed ensembled temperature scaling to jointly optimize accuracy and calibration.

## 3 PROBLEM DESCRIPTION

### 3.1 UNCERTAINTY CALIBRATION ON GNN

This paper addresses the calibration of common semi-supervised node classification tasks. Let $\mathcal{G} = (\mathcal{V}, \mathcal{E})$ represent the graph, where $\mathcal{V}$ is the set of nodes and $\mathcal{E}$ is the set of edges. The adjacency matrix is denoted by $\mathbf{A} \in \mathbb{R}^{N \times N}$, where $N = |\mathcal{V}|$ represents the number of nodes. Let $\mathcal{X}$ be the input space and $\mathcal{Y}$ be the label space, with $y_i \in \mathcal{Y} = \{1, \ldots, K\}$, where $K \geq 2$ is the number of classes. Let $\mathbf{X} \in \mathcal{X}$ and $\mathbf{Y} \in \mathcal{Y}$ denote the input features and labels, respectively. A subset of

nodes with ground-truth labels, $\mathcal{L} \subset \mathcal{V}$, is selected for training. The goal of semi-supervised node classification is to infer the labels of the unlabeled nodes, $\mathcal{U} = \mathcal{V} \backslash \mathcal{L}$. The node-wise input feature matrix and the label vector are denoted by $\mathbf{X} = [\mathbf{x}_1, \ldots, \mathbf{x}_N]^\top$ and $\mathbf{Y}$, respectively.

A GNN model $f_\theta$ is trained to address this problem by considering the node-wise features $\{\mathbf{x}_i\}_{i:v_i \in \mathcal{L}}$ and the adjacency matrix $\mathbf{A}$, where $\theta$ represents the learnable parameters. For a given node $i$, the logit output of the GNN is represented as $\mathbf{z}_i = f_\theta(\mathbf{x}_i, \mathbf{A}) = [z_{i,1}, \ldots, z_{i,K}]^\top$, where $K$ is the total number of classes. The predicted label for node $i$ is given by $\hat{y}_i = \arg\max_{k \in [1,K]} z_{i,k}$, selecting the class with the highest logit value. The confidence or predicted probability for node $i$ is then defined as $\hat{p}_i = \max_{k \in [1,K]} \text{softmax}(z_{i,k})$, where $\text{softmax}(z_{i,k})$ transforms the logits into probability scores for each class $k$. We define $f_\theta$ to be perfectly calibrated as (Guo et al., 2017; Chung et al., 2021):

$$\mathbb{P}(\hat{y}_i = y_i \mid \hat{p}_i = p) = p, \quad \forall p \in [0,1]. \tag{1}$$

The calibration error (CE) for a graph neural network $f_\theta$ can be defined as the discrepancy between the predicted probability $\hat{p}_i = \max_k \text{softmax}(z_{i,k})$ and the true probability. For node $i$, the calibration error is expressed as:

$$\text{CE}(f_\theta) = \mathbb{E}\left[|\hat{p}_i - \mathbb{P}(Y = \hat{y}_i \mid \hat{p}_i)|\right]. \tag{2}$$

If $\text{CE}(f_\theta) = 0$, then $f_\theta$ is perfectly calibrated. However, direct computation of the calibration error from the model outputs is challenging in practice. Therefore, the ECE is proposed to approximate CE by discretization (binning) and summation, where the model outputs are partitioned into $B$ intervals (bins) with binning strategy $\mathcal{B} = \{I_1, I_2, \ldots, I_B\}$. A common binning strategy $\mathcal{B}^0$ divides bins by sorting confidence scores $\hat{p}_i$ in ascending order, ensuring each bin $I_j^0$ contains approximately the same number of nodes, i.e., $|I_j^0| \approx \frac{N}{B}$.

The ECE is defined as the weighted average of the absolute difference between the accuracy and confidence in each bin, denoted by $Acc(I_j)$ and $Conf(I_j)$:

$$\text{ECE}(f_\theta, \mathcal{B}) = \sum_{j=1}^{B} \frac{|I_j|}{N} |Acc(I_j) - Conf(I_j)| = \sum_{j=1}^{B} \frac{|I_j|}{N} \left| \mathbb{E}[Y = \hat{y}_i \mid \hat{p}_i \in I_j] - \frac{1}{|I_j|} \sum_{i \in I_j} \hat{p}_i \right|. \tag{3}$$

Importantly, $ECE(f_\theta, \mathcal{B}) \leq CE(f_\theta)$ for any binning strategy, and this approximation is widely used for estimating the overall calibration error (Kumar et al., 2019). By default, we evaluate models with $\text{ECE}(f_\theta, \mathcal{B}^0)$.

### 3.2 Node-wise Parametric Calibration

TS is the classical parametric calibration approach, which adjusts model confidence by scaling logits with a temperature $T > 0$ before softmax, smoothing outputs without affecting accuracy. The temperature $T$ controls the smoothness of predicted probabilities: $T = 1$ uses logits directly, while increasing $T$ spreads probabilities, improving calibration in overconfident models.

Recent studies extend this by learning node-wise temperatures for each node $i$ in graph classification (Wang et al., 2021; Hsu et al., 2022). Commonly, node-wise temperature scaling for calibration is conducted by applying an extra GNN layer to learn the node-wise optimal smoothing parameter $T_i$ and do the following operation for the logits:

$$z_i' = z_i / T_i. \tag{4}$$

We should mention that in training the smoothing temperature $T$, the loss function is always the negative log-likelihood function, which is categorized to the proper score (Gruber & Buettner, 2022) in the classification task. The famous ECE error can not be utilized as a training objective function for their discontinuity. Moreover, these post-hoc calibration methods are usually trained on the validation set. We follow these training settings in our main context.

## 4 GETS

### 4.1 Ensemble Strategy Meets Node-wise Calibration

Ensemble strategies are widely recognized for improving model robustness and calibration by aggregating predictions from multiple models. For example, in the context of ensemble temperature

scaling (ETS) (Zhang et al., 2020), each model learns its own temperature parameter $T_m$ to calibrate its predictions. The ensemble strategy combines the calibrated outputs from multiple calibration models, averaging their predictions to produce a more reliable and better-calibrated result. The ensemble weighting is formulated as a weighted sum over temperature-scaled logits (Zhang et al., 2020): $T(z) = \sum_{m=1}^{M} w_m \cdot t_m(z)$, where $w_m$ are non-negative coefficients summing to 1, and $t_m(z)$ represents the value of temperature of the $m$-th model TS model trained on logit $z$, with $M$ being the number of models.

The ensemble inherits the accuracy-preserving properties from its individual components, with the weights $w_m$ controlling the contribution of each calibrated prediction. While ETS leverages model diversity to mitigate overconfidence and improve calibration quality, it falls short in handling the intricate structural problems inherent in graph data. Specifically, ETS considers only the temperatures $t_m(z)$, logit $z$, and bin size $B$ to optimize the weights, without incorporating graph structural information such as node degrees. Moreover, the learned temperature is a single value uniformly applied across all nodes, which cannot adapt to the node-wise calibration required in graphs.

We propose to learn a node-wise ensemble strategy for the GNN calibration task. To incorporate the different sources of information, we ensemble based on not only the calibration models but also diverse inputs, including logits $z_i$, node feature $\mathbf{x}_i$ and node degrees $d_i$, as well as their combination as the calibration model inputs. We use $g_m$ to represent the $m$-th GNN calibration model that outputs the calibrated logits. Due to the monotone-preserving property of TS, the weighted summation of the logits still keeps the invariant accuracy output. For each node $i$, the calibrated logits $z'_i$ are computed as a weighted sum of the outputs from multiple calibration models:

$$z'_i = \sum_{m=1}^{M} w_{i,m} \cdot g_m(z_i, \mathbf{x}_i, d_i; \theta_m), \tag{5}$$

where $w_{i,m} \geq 0$ are the node-specific ensemble weights for node $i$ and model $m$, satisfying $\sum_{m=1}^{M} w_{i,m} = 1$. We further point out that in our expert setting, the input of each expert only contains one element in $\mathbf{x}_i, d_i, z_i$. This gives reasonable learning tasks for different experts and rules out the problem arising in concatenating vectors of different input spaces. Each calibration model $g_m$ can focus on different aspects of the calibration process by utilizing various combinations of inputs. For example, one model may calibrate based solely on the logits, another may incorporate node features, and another may consider the node degrees. Specifically, $w_{i,m}$ can therefore be determined by a gating function $G(\cdot)$ that outputs a probability distribution over the calibration models for each node:

$$[w_{i,1}, w_{i,2}, \ldots, w_{i,M}] = G(\{g(z_i, \mathbf{x}_i, d_i; \theta_m)\}_{m=1}^{M}; \mathbf{W}_g, \mathbf{W}_n), \tag{6}$$

where $\mathbf{W}_g$ and $\mathbf{W}_n$ represent the parameters of the gating function $G(\cdot)$, which will be introduced later on. We ensemble the feature inputs by concatenating different combinations of inputs into $g_m(\cdot)$. The key component that remains unknown is how we should do the information aggregation and obtain the gating function to select models.

We include logits $z$, node features $\mathbf{x}$, and degree (embeddings) $d$ to capture key factors influencing calibration. Including logits is crucial because they contain the raw prediction information before applying softmax, reflecting the model's confidence. Incorporating node features allows the model to address individual calibration needs, as feature-based methods binning methods have successfully captured unique sample point characteristics in calibration task (Huang et al., 2022). Adding degree embeddings tackles structural disparities since our experiments show that degree imbalance leads to varying calibration errors among nodes. Note that degrees are the foundation for graph structure information but we also explore other structural features as inputs in Appendix A.9. By combining these inputs, our ensemble leverages multiple information sources, resulting in a more robust and effective calibration strategy.

## 4.2 Ensemble Calibration by Graph MoE

The node-wise ensemble idea mentioned in the last section coincides with the so-called MoE framework (Jordan & Jacobs, 1994; Jacobs et al., 1991; Jacobs, 1995; Wang et al., 2024). MoE allows for specialization by assigning different experts to handle distinct inputs, enabling each expert to focus on specific factors. This specialization is particularly beneficial when inputs are diverse, as it allows experts to learn the unique properties of each factor more effectively than one single GAT model

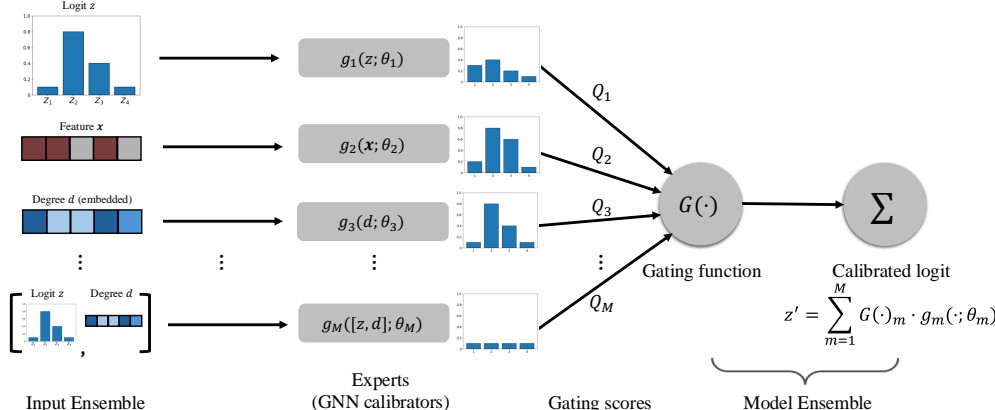

Figure 2: Illustration of input and model ensemble calibration. The input ensemble explores different combinations of input types, while the model ensemble employs a MoE framework to select the most effective experts for calibration. The final calibrated outputs are weighted averages determined by the gating mechanism. The notation $(\cdot)$ indicates different input types for the function.

(Hsu et al., 2022) for many inputs at the same time. Moreover, MoE employs sparse activation through a gating network, which selectively activates only relevant experts for each input, reducing computational overhead compared to the dense computation in GATs (Shazeer et al., 2017; Lepikhin et al., 2020). The overall framework design is shown in Figure 2 (Zhang et al., 2020). To adapt MoE for graph-based calibration tasks, we propose the **GETS** framework that integrates both the input ensemble and the model ensembles. In GNNs like GCNs, nodes update their representations by aggregating information from their neighbors. The standard GCN propagation rule is (Wang et al., 2024):

$$\mathbf{h}_i' = \sigma \left( \sum_{j \in \mathcal{N}_i} \frac{1}{\sqrt{|d_i||d_j|}} \mathbf{h}_j \mathbf{W} \right), \tag{7}$$

where $\mathbf{h}_i'$ is the updated feature vector for node $i$ after the GCN layer, $d_i$ is the set of neighboring nodes of node $i$. We train different GCNs based on various ensemble of inputs as different experts, $\mathbf{d}_j$ is the feature vector of the neighboring node $j$, and $\mathbf{W}$ is a learnable matrix for the GCN network. $\frac{1}{\sqrt{|d_i||d_j|}}$ is the normalization term where $|d_i|$ and $|d_j|$ are the degrees (number of neighbors) of nodes $i$ and $j$, respectively. In many GNN papers, the normalization term is also denoted by $\tilde{\mathbf{D}}^{-1/2} \tilde{\mathbf{A}} \tilde{\mathbf{D}}^{-1/2}$ with $\tilde{\mathbf{A}} = \mathbf{A} + \mathbf{I}$ is the adjacency matrix with added self-loops. $\tilde{\mathbf{D}}$ is the corresponding degree matrix with $\tilde{\mathbf{D}}_{ii} = \sum_j \tilde{\mathbf{A}}_{ij}$ and $\sigma$ is a non-linear activation function.

While GCNs aggregate neighbor information effectively, they apply the same transformation to all nodes, which may not capture diverse patterns or complex dependencies in the graph. Following our aforementioned notations, the Graph MoE framework introduces multiple experts by learning different GNN models with a gating mechanism for aggregation. The propagation rule in Graph MoE training is:

$$\mathbf{h}_i' = \sigma \left( \sum_{m=1}^{M} G(\mathbf{h}_i)_m \cdot g_m \left( \mathbf{h}_j; \theta_m \right) \right), \tag{8}$$

where, for consistency with Equation 5, $g_m(\cdot)$ represents the $m$-th GNN expert function that processes a specific ensemble of inputs—such as logits $z_i$, node features $\mathbf{x}_i$, and node degrees. The parameter $\theta_m$, associated with expert $m$, is analogous to the weight matrix $\mathbf{W}_m$ from Equation 7 (we omit the message-passing aggregation for clarity). The gating mechanism determines the contribution of each expert for node $i$ is

$$G(\mathbf{h}_i) = \text{Softmax} \left( \text{TopK}(Q(\mathbf{h}_i), k) \right), \tag{9}$$

with the function, TopK to select the best $k$ experts with gating scores $Q(\mathbf{h}_i)$. $G(\mathbf{h}_i)_m$ is the gating function outputted weights for $m$-th expert, with $\mathbf{h}_i$ usually being the last layer output of $m$-th expert. The gating score $Q(\mathbf{h}_i)$ is computed as:

$$Q(\mathbf{h}_i) = \mathbf{h}_i \mathbf{W}_g + \epsilon \cdot \text{Softplus}(\mathbf{h}_i \mathbf{W}_n), \tag{10}$$

where $\mathbf{W}_g$ and $\mathbf{W}_n$ are learnable weight matrices that control clean and noisy scores, respectively. Noisy gating $\epsilon \sim \mathcal{N}(0,1)$ introduces Gaussian noise for exploration. In the context of calibration, we adapt GETS to combine multiple calibration experts, each specialized in handling specific calibration factors. For those unselected models the gating function $G(\cdot)$ sets them to zeroes to be consistent with Equation 9, for example, if we select the best 2 experts, the rest $M-2$ outputs, the gating function $G(\cdot)$ assigns them to 0. Finally, for each node $i$, the calibrated logits $z_i'$ are computed as, in a similar form as Equation 5:

$$z_i' = \sum_{m=1}^{M} G(\mathbf{h}_i)_m \cdot g_m(z_i; \theta_m), \tag{11}$$

Each calibration expert $g_m$ can implement different node-wise calibration strategies, where the model usually includes node-wise temperatures. The model is trained by minimizing a calibration loss over all nodes: $L = \frac{1}{N} \sum_{i=1}^{N} \ell_{\text{cal}}(z_i', y_i)$, where $\ell_{\text{cal}}$ is a differentiable calibration loss function (e.g., cross-entropy loss) mentioned before and $N$ is the number of nodes. By leveraging GETS, we enable the calibration model to specialize in handling diverse calibration factors via different experts and adaptively select the most relevant experts for each node through the gating mechanism using node features and topology.

## 5 EXPERIMENTS

### 5.1 EXPERIMENTAL SETUP

We include the 10 commonly used graph classification networks for a thorough evaluation, The data summary is given in Table 1, refer to Appendix A.2 for their sources. The train-val-test split is 20-10-70 (Hsu et al., 2022; Tang et al., 2024), note that uncertainty calibration models are trained on the validation set, which is also referred to as the calibration set. We randomly generate 10 different splits of training, validation, and testing inputs and run the models 10 times on different splits.

For the base classification GNN model, i.e., the uncalibrated model $f_\theta$, we choose the vanilla GCN (Kipf & Welling, 2016), GAT (Veličković et al., 2017), and GIN (Xu et al., 2018). We tune the vanilla models to get optimal classification performance, see Appendix A.1.1 for more details about the parameters. After training, we evaluate models by ECE with $B = 10$ equally sized bins.

Table 1: Summary of datasets. The average degree is defined as $\frac{|2\mathcal{E}|}{|\mathcal{V}|}$ to measure the connectivity of the network.

|  | Citeseer | Computers | Cora | Cora-full | CS | Ogbn-arxiv | Photo | Physics | Pubmed | Reddit |
|---|---|---|---|---|---|---|---|---|---|---|
| #Nodes | 3,327 | 13,381 | 2,708 | 18,800 | 18,333 | 169,343 | 7,487 | 34,493 | 19,717 | 232,965 |
| #Edges | 12,431 | 491,556 | 13,264 | 144,170 | 163,788 | 2,501,829 | 238,087 | 495,924 | 108,365 | 114,848,857 |
| Avg. Degree | 7.4 | 73.4 | 9.7 | 15.3 | 17.8 | 29.5 | 63.6 | 28.7 | 10.9 | 98.5 |
| #Features | 3,703 | 767 | 1,433 | 8,710 | 6,805 | 128 | 745 | 8,415 | 500 | 602 |
| #Classes | 6 | 10 | 7 | 70 | 15 | 40 | 8 | 5 | 3 | 41 |

All our experiments are implemented on a machine with Ubuntu 22.04, with 2 AMD EPYC 9754 128-Core Processors, 1TB RAM, and 10 NVIDIA L40S 48GB GPUs.

### 5.2 CONFIDENCE CALIBRATION EVALUATION

**Baselines.** For comparison, we evaluate our approach against several baseline models, including both classic parametric post-hoc calibration methods and graph calibration techniques. The classic methods include TS (Guo et al., 2017; Kull et al., 2019), Vector Scaling (VS) (Guo et al., 2017), and ETS (Zhang et al., 2020). These baselines are primarily designed for standard i.i.d. multi-class classification.

Table 2: Calibration performance across datasets evaluated by ECE. Results are reported as mean ± standard deviation over 10 runs. 'Uncal.' denotes uncalibrated results, and 'OOM' indicates out-of-memory issues where the model could not be run. The best results for each dataset are marked bold.

| Dataset | Classifier | Uncal. | TS | VS | ETS | CaGCN | GATS | GETS |
|---|---|---|---|---|---|---|---|---|
| **Citeseer** | GCN | 20.51 ± 2.94 | 3.10 ± 0.34 | 2.82 ± 0.56 | 3.06 ± 0.33 | 2.79 ± 0.37 | 3.36 ± 0.68 | **2.50 ± 1.42** |
| | GAT | 16.06 ± 0.62 | 2.18 ± 0.20 | 2.89 ± 0.28 | 2.38 ± 0.25 | 2.49 ± 0.47 | 2.50 ± 0.36 | **1.98 ± 0.30** |
| | GIN | 4.12 ± 2.83 | 2.39 ± 0.32 | 2.55 ± 0.34 | 2.58 ± 0.34 | 2.96 ± 1.49 | 2.03 ± 0.23 | **1.86 ± 0.22** |
| **Computers** | GCN | 5.63 ± 1.06 | 3.29 ± 0.48 | 2.80 ± 0.29 | 3.45 ± 0.56 | **1.90 ± 0.42** | 3.28 ± 0.52 | 2.03 ± 0.35 |
| | GAT | 6.71 ± 1.57 | 1.83 ± 0.27 | 2.19 ± 0.15 | 1.89 ± 0.24 | 1.88 ± 0.36 | 1.89 ± 0.34 | **1.77 ± 0.27** |
| | GIN | 4.00 ± 2.84 | 3.92 ± 2.59 | 2.57 ± 1.06 | 2.91 ± 2.07 | **2.37 ± 1.87** | 3.34 ± 2.08 | 3.14 ± 3.70 |
| **Cora** | GCN | 22.62 ± 0.84 | 2.31 ± 0.53 | 2.43 ± 0.41 | 2.56 ± 0.47 | 3.22 ± 0.81 | 2.60 ± 0.76 | **2.29 ± 0.52** |
| | GAT | 16.70 ± 0.75 | 1.82 ± 0.39 | 1.71 ± 0.28 | **1.67 ± 0.33** | 2.85 ± 0.66 | 2.58 ± 0.71 | 2.00 ± 0.48 |
| | GIN | 3.59 ± 0.65 | 2.51 ± 0.33 | **2.20 ± 0.38** | 2.29 ± 0.30 | 2.79 ± 0.26 | 2.64 ± 0.39 | 2.63 ± 0.87 |
| **Cora-full** | GCN | 27.73 ± 0.22 | 4.43 ± 0.10 | 3.35 ± 0.22 | 4.34 ± 0.09 | 4.08 ± 0.25 | 4.46 ± 0.17 | **3.32 ± 1.24** |
| | GAT | 37.51 ± 0.22 | 2.27 ± 0.33 | 3.55 ± 0.13 | **1.37 ± 0.25** | 3.96 ± 1.15 | 2.47 ± 0.35 | 1.52 ± 2.27 |
| | GIN | 10.69 ± 3.55 | 3.33 ± 0.22 | 2.43 ± 0.19 | 2.18 ± 0.16 | 4.74 ± 0.66 | 3.98 ± 0.35 | **1.95 ± 0.37** |
| **CS** | GCN | 1.50 ± 0.10 | 1.49 ± 0.07 | 1.39 ± 0.10 | 1.44 ± 0.08 | 2.00 ± 0.98 | 1.51 ± 0.10 | **1.34 ± 0.10** |
| | GAT | 4.30 ± 0.42 | 2.98 ± 0.15 | 3.02 ± 0.18 | 2.98 ± 0.15 | 2.57 ± 0.38 | 2.56 ± 0.15 | **1.05 ± 0.27** |
| | GIN | 4.36 ± 0.31 | 4.31 ± 0.31 | 4.28 ± 0.28 | 3.04 ± 1.19 | 1.81 ± 1.09 | 1.24 ± 0.45 | **1.15 ± 0.32** |
| **Ogbn-arxiv** | GCN | 9.90 ± 0.77 | 9.17 ± 1.10 | 9.29 ± 0.88 | 9.60 ± 0.92 | 2.32 ± 0.33 | 2.97 ± 0.34 | **1.85 ± 0.22** |
| | GAT | 6.90 ± 0.47 | 3.56 ± 0.10 | 4.39 ± 0.08 | 3.55 ± 0.10 | 3.52 ± 0.13 | 3.90 ± 0.77 | **2.34 ± 0.12** |
| | GIN | 11.56 ± 0.27 | 11.35 ± 0.27 | 11.26 ± 0.43 | 11.38 ± 0.40 | 6.01 ± 0.28 | 6.41 ± 0.29 | **3.33 ± 0.36** |
| **Photo** | GCN | 4.05 ± 0.42 | 2.14 ± 0.46 | 2.11 ± 0.23 | 2.21 ± 0.34 | 2.42 ± 0.76 | 2.24 ± 0.30 | **2.07 ± 0.31** |
| | GAT | 4.97 ± 0.75 | 2.06 ± 0.57 | 1.71 ± 0.23 | 2.57 ± 0.59 | 1.69 ± 0.14 | 2.05 ± 0.45 | **1.10 ± 0.25** |
| | GIN | 3.37 ± 2.40 | 3.12 ± 1.85 | 3.77 ± 1.10 | 3.34 ± 1.70 | 2.42 ± 1.90 | 3.37 ± 1.85 | **1.43 ± 0.71** |
| **Physics** | GCN | 0.99 ± 0.10 | 0.96 ± 0.05 | 0.96 ± 0.05 | 0.87 ± 0.05 | 1.34 ± 0.45 | 0.91 ± 0.04 | **0.87 ± 0.09** |
| | GAT | 1.52 ± 0.29 | 0.37 ± 0.10 | 0.44 ± 0.05 | 0.47 ± 0.22 | 0.69 ± 0.11 | 0.48 ± 0.23 | **0.29 ± 0.05** |
| | GIN | 2.11 ± 0.14 | 2.08 ± 0.14 | 2.08 ± 0.14 | 1.64 ± 0.09 | 2.36 ± 0.52 | 0.90 ± 0.66 | **0.44 ± 0.10** |
| **Pubmed** | GCN | 13.26 ± 1.20 | 2.45 ± 0.30 | 2.34 ± 0.27 | 2.05 ± 0.26 | 1.93 ± 0.36 | 2.19 ± 0.27 | **1.90 ± 0.40** |
| | GAT | 9.84 ± 0.16 | 0.80 ± 0.07 | 0.95 ± 0.08 | 0.81 ± 0.09 | 1.04 ± 0.08 | 0.81 ± 0.07 | **0.78 ± 0.15** |
| | GIN | 1.43 ± 0.15 | 1.01 ± 0.06 | 1.04 ± 0.06 | 1.00 ± 0.04 | 1.40 ± 0.47 | 1.00 ± 0.06 | **0.92 ± 0.13** |
| **Reddit** | GCN | 6.97 ± 0.10 | 1.72 ± 0.07 | 2.02 ± 0.07 | 1.72 ± 0.07 | 1.50 ± 0.08 | OOM | **1.49 ± 0.07** |
| | GAT | 4.75 ± 0.15 | 3.26 ± 0.08 | 3.43 ± 0.10 | 3.52 ± 0.10 | 0.81 ± 0.09 | OOM | **0.62 ± 0.08** |
| | GIN | 3.22 ± 0.09 | 3.17 ± 0.13 | 3.19 ± 0.10 | 3.25 ± 0.14 | 1.63 ± 0.21 | OOM | **1.57 ± 0.12** |

In addition, we compare against graph-based calibration models that utilize a two-layer GNN to learn temperature values for each node. These include Graph convolution network as a calibration function (CaGCN) (Wang et al., 2021) and Graph Attention Temperature Scaling (GATS) (Hsu et al., 2022), which use GCN or GAT layers to process logits and incorporate graph-specific information for calibration. All models are tuned to their optimal performance, with detailed parameter settings provided in Appendix A.1.3.

**GETS settings.** For the parameters of GETS, we constructed input ensembles for each node $i$ as $\{z_i, \mathbf{x}_i, d_i, [z_i, \mathbf{x}_i], [\mathbf{x}_i, d_i], [z_i, d_i], [z_i, \mathbf{x}_i, d_i]\}$. We trained $M = 7$ experts and selected the top 2 experts based on their gating scores, as described in Equation 9. For the degree input $d_i$, we mapped the integer degree of each node into a dense vector of fixed size 16 using `torch.nn.Embedding`, before training the experts with this representation.

After parameter tuning, we train the GNN expert $g_m(\cdot), m = 1, \ldots, M$ for 1000 epochs with a patience of 50 across all datasets. The calibration learning rate is set to 0.1, except for the *Reddit* dataset, which uses a rate of 0.01. Weight decay is set to 0 for most datasets, except for *Citeseer* (0.01) and *Reddit* (0.001). By default, we use a two-layer GCN to train an expert and apply noisy gating with $\epsilon \sim \mathcal{N}(0, 1)$. We also report the ECE for uncalibrated predictions as a reference.

For all experiments, the pre-trained GNN classifiers are frozen, and the predicted logits $z$ from the validation set are fed into our calibration model as inputs. Further comparison settings and hyperparameters are detailed in the Appendix A.1.2.

**Results.** Table 2 presents the calibration results evaluated by ECE, showing that our proposed method, GETS, consistently achieves superior performance across various classifiers and datasets. On average, GETS improves ECE by **28.60%** over CaGCN, **26.62%** over GATS, and **28.09%** over ETS across all datasets. While GETS demonstrates significant improvements on most datasets,

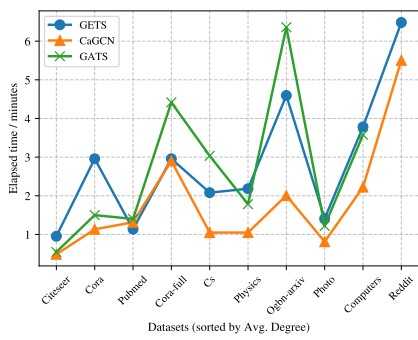

(a) Elapsed time for 10 runs.

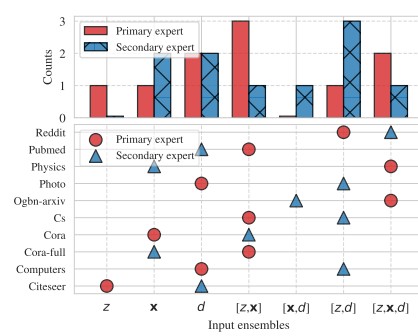

(b) Input ensemble selected by sparse activation of experts.

Figure 3: Illustration of computational efficiency and expert selection properties. (a): Elapsed time for training each model for 10 runs; (b): Primary and secondary expert selections across datasets for various input ensembles. The top bar plot shows the frequency of expert selection, highlighting the significance of combining logits and features in calibration across datasets.

some exceptions are noted in *Computers*, *Cora*, and *Cora-full*, where the gains are less pronounced or slightly negative.

Interestingly, the best results often come from models that incorporate ensemble strategies, such as ETS, highlighting the importance of combining these techniques for effective calibration. This underscores the value of our GETS framework, which leverages both ensemble learning and GNN structures to achieve node-specific calibration.

Moreover, GETS proves to be more scalable than GATS, which suffers from out-of-memory (OOM) issues in larger networks like *Reddit*. These results affirm the robustness and scalability of GETS, showcasing its ability to handle both small and large-scale datasets while significantly reducing calibration error. Overall, GETS demonstrates its effectiveness in addressing node-wise calibration by incorporating input and model ensemble, offering clear improvements over other GNN calibration methods like CaGCN and GATS.

## 5.3 TIME COMPLEXITY

CaGCN uses a two-layer GCN with a total time complexity of $O(2(|\mathcal{E}|F + |\mathcal{V}|F^2))$, where $F$ is the feature dimension (Blakely et al., 2021). GATS adds attention mechanisms, leading to a complexity of $O(2(|\mathcal{E}|FH + |\mathcal{V}|F^2))$, with $H$ as the number of attention heads (Veličković et al., 2017). GETS, fundamentally Graph MoE, introduces additional complexity by selecting the top $k$ experts per node, yielding $O(k(|\mathcal{E}|F + |\mathcal{V}|F^2) + |\mathcal{V}|MF)$, where $k \ll M$. This reduces computational costs compared to using all experts, making GETS scalable while maintaining competitive performance. Training multiple experts might make GETS less time-efficient than its counterpart. However, GETS scales linearly with the number of nodes, and selecting lightweight models like GCN as experts helps manage the time complexity effectively. In practice, the GETS model is computationally efficient, as shown in Figure 3a, even though multiple experts are trained, GETS remains stable elapsed time comparably efficient with the other two models that are trained with a single GNN model.

## 5.4 EXPERT SELECTION

One advantage of GETS is its ability to visualize which experts, trained on specific input types, are most important for each node. We illustrate this by coloring each node according to the selected expert for it. As sparse activation in the gating function highlights the significance of different input ensembles, we present the top two expert selections (primary and secondary) for each dataset in Figure 3b. Gating results are averaged over 10 runs to select the primary and secondary experts.

The results show a diverse range of expert choices, with the combination of logits and feature $[z, x]$ frequently chosen as the primary experts, particularly for complex datasets like *Ogbn-arxiv* and *Reddit*. In contrast, degree-based experts ($d$) are often selected as secondary. For smaller datasets such as *Cora* and *Pubmed*, individual expert selections are more prevalent. Note that only using logits $z$ as the inputs is not preferred by the gating function, which indicates the importance of ensembling

different inputs. These findings indicate that integrating multiple input types—especially logits and features—improves calibration, with the optimal expert combination depending on the complexity of the dataset.

## 5.5 ABLATION STUDIES

In this section, we evaluate the impact of the backbone calibration model in our proposed GETS framework. We investigate whether the choice of GNN backbone (e.g., GCN, GAT, GIN) significantly affects calibration performance, offering insights into the robustness of GETS across different architectures. We reuse the parameters tuned for default GETS. For the ablation discussion that follows, we use GCN as the default uncalibrated model.

Table 3: Ablation studies on different expert models, measured by ECE.

| Expert | Citeseer | Computers | Cora | Cora-full | CS | Ogbn-arxiv | Photo | Physics | Pubmed | Reddit |
|---|---|---|---|---|---|---|---|---|---|---|
| GETS-GAT | 4.09 ± 0.71 | 3.64 ± 1.94 | 2.96 ± 0.70 | 14.04 ± 5.70 | 4.91 ± 3.93 | 1.61 ± 0.28 | 3.43 ± 1.82 | 2.57 ± 2.23 | 1.96 ± 0.59 | OOM |
| GETS-GIN | 4.34 ± 1.36 | 4.56 ± 3.33 | 5.53 ± 0.59 | 2.83 ± 0.46 | 2.29 ± 0.82 | 2.48 ± 0.30 | 4.06 ± 2.96 | 1.16 ± 0.14 | 2.30 ± 0.58 | 4.64 ± 1.03 |

Generally, choosing GAT and GIN as the models to train experts does not provide significant advantages over using GCN, as shown in Table 3. GETS-GAT yields comparable performance to GATS, and also encounters the same OOM issue in the *Reddit* dataset, demonstrating its limitations in handling large-scale datasets. Furthermore, while GETS-GIN shows improved results on datasets like *Cora-full*, it underperforms on several other datasets, such as *Citeseer*, *Computers*, and *Reddit*, compared to GCN-based calibration. Notably, in simpler datasets or those with smaller graph structures, GCN tends to achieve better calibration results without overfitting. This suggests that a simpler GNN-base calibrator, like GCN, is preferable for training experts as it offers a more balanced trade-off between model complexity and calibration performance, while avoiding potential issues like overfitting, which may arise when more complex GNN architectures are used.

## 6 CONCLUSION

We propose the GETS framework that combines input and model ensemble strategies within a Graph MoE architecture. It effectively incorporates diverse inputs — logits, node features, and degree embeddings, as well as using a sparse gating mechanism to adaptively select the most relevant experts for each node. This mechanism enhances node-wise calibration performance in general GNN classification tasks.

Our extensive experiments on 10 benchmark GNN datasets demonstrated that GETS consistently outperforms state-of-the-art calibration methods, reducing the expected calibration error significantly across different datasets and GNN architectures. GETS also proved to be computationally efficient and scalable, effectively handling large-scale graphs without significant overhead. By integrating multiple influential factors and leveraging ensemble strategies, GETS enhances the reliability and trustworthiness of GNN prediction results.

### ACKNOWLEDGMENTS

Dingyi Zhuang acknowledges the support by the U.S. Department of Energy's Office of Energy Efficiency and Renewable Energy (EERE) under the Vehicle Technology Program Award Number DE-EE0009211. Yunhan Zheng acknowledges the support by the National Research Foundation (NRF), Prime Minister's Office, Singapore under its Campus for Research Excellence and Technological Enterprise (CREATE) programme. The Mens, Manus, and Machina (M3S) is an interdisciplinary research group (IRG) of the Singapore MIT Alliance for Research and Technology (SMART) centre. Shenhao Wang acknowledges the support of Research Opportunity Seed Fund (ROSF2023) from the University of Florida.

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

# A APPENDIX

## A.1 MODEL DETAILS

### A.1.1 BASE GNN CLASSIFIER SETTINGS

For the base GNN classification model (i.e., the uncalibrated model), we follow the architecture and parameter setup outlined by Kipf & Welling (2016); Veličković et al. (2017); Xu et al. (2018), with modifications to achieve optimal performance. Specifically, we use a two-layer GCN, GAT, or GIN model and tune the hidden dimension from the set $\{16, 32, 64\}$. We experiment with dropout rates ranging from 0.5 to 1, and we do not apply any additional normalization.

During training, we use a learning rate of $1 \times 10^{-2}$. We tune the weight decay parameter to prevent overfitting and consider adding early stopping with patience of 50 epochs. The model is trained for a maximum of 200 epochs to ensure convergence. The specifics are summarized in Table 4.

Table 4: Summary of GNN and Training Parameters

| Dataset | Num Layers | Hidden Dimension | Dropout | Training Epochs | Learning Rate | Weight Decay |
|---|---|---|---|---|---|---|
| Citeseer | 2 | 16 | 0.5 | 200 | 1e-2 | 5e-4 |
| Computers | 2 | 64 | 0.8 | 200 | 1e-2 | 1e-3 |
| Cora-Full | 2 | 64 | 0.8 | 200 | 1e-2 | 1e-3 |
| Cora | 2 | 16 | 0.5 | 200 | 1e-2 | 5e-4 |
| CS | 2 | 64 | 0.8 | 200 | 1e-2 | 1e-3 |
| Ogbn-arxiv | 2 | 256 | 0.5 | 200 | 1e-2 | 0 |
| Photo | 2 | 64 | 0.8 | 200 | 1e-2 | 1e-3 |
| Physics | 2 | 64 | 0.8 | 200 | 1e-2 | 1e-3 |
| Pubmed | 2 | 16 | 0.5 | 200 | 1e-2 | 5e-4 |
| Reddit | 2 | 16 | 0.5 | 200 | 1e-2 | 5e-4 |

### A.1.2 GETS PARAMETER SETTINGS

The backbone GNN models used for calibration are identical to the base classification models, ensuring consistent feature extraction and processing. We tune the hidden dimension for the calibrators from the set $\{16, 32, 64\}$, depending on the dataset's complexity, while dropout rates are chosen between 0.2 and 0.8 to balance regularization and model performance.

GETS includes an expert selection mechanism that incorporates logits, features, and node degree information, either individually or in combination, depending on the configuration that best fits the dataset. For all experiments, we maintain a fixed bin size of 10 for uncertainty estimation. The calibrators are trained with a learning rate of either $1 \times 10^{-1}$ or $1 \times 10^{-3}$, depending on the model's complexity, and we tune weight decay ranging from 0 to 0.1 to regularize the training process further. The calibrators are trained for up to 1000 epochs, with early stopping applied based on patience of 50 epochs.

This comprehensive setup ensures robust calibration of the GNN models across diverse datasets. The full details of the calibration parameters are summarized in Table 5.

Table 5: Summary of GETS Parameters Across Datasets

| Dataset | Hidden Dim | Dropout | Num Layers | Learning Rate | Weight Decay |
|---|---|---|---|---|---|
| Citeseer | 16 | 0.2 | 2 | 0.1 | 0.01 |
| Computers | 16 | 0.8 | 2 | 0.1 | 0 |
| Cora-Full | 16 | 0.8 | 2 | 0.1 | 0 |
| Cora | 16 | 0.5 | 2 | 0.001 | 0 |
| CS | 16 | 0.8 | 2 | 0.1 | 0 |
| Ogbn-arxiv | 16 | 0.2 | 2 | 0.1 | 0.01 |
| Photo | 32 | 0.8 | 2 | 0.1 | 0.001 |
| Physics | 16 | 0.2 | 2 | 0.1 | 0.001 |
| Pubmed | 16 | 0.8 | 2 | 0.1 | 0 |
| Reddit | 64 | 0.8 | 2 | 0.01 | 0.001 |

### A.1.3 BASELINE MODELS SETTINGS

For all the baseline calibration models, we ensure consistent training and calibration settings. The calibration process is set to run for a maximum of 1000 epochs with early stopping applied after 50 epochs of no improvement. The calibration learning rate is set to 0.01, with no weight decay (0) to avoid regularization effects during calibration. We use 10 bins for the calibration process, and the default calibration method is chosen as Vector Scaling (VS). For the GATS model, we apply two attention heads and include bias terms. Additionally, we apply a dropout rate of 0.5 during the calibration to prevent overfitting, ensuring robustness across different setups. Full implementations can be also found on our GitHub.

**TS** (Guo et al., 2017) adjusts the logits output by a model using a single scalar temperature parameter $T > 0$. The calibrated logits $z_i'$ for sample $i$ are computed as:

$$z_i' = \frac{z_i}{T} \tag{12}$$

The temperature $T$ is learned by minimizing the negative log-likelihood (NLL) loss on the validation (calibration) set:

$$\min_{T>0} \mathcal{L}_{\text{NLL}}(T) = -\frac{1}{N} \sum i = 1^N \log \left( \text{softmax} \left( \frac{z_i}{T} \right)_{y_i} \right) \tag{13}$$

where $y_i \in \{1, \ldots, K\}$ is the true label for sample $i$, $N$ is the number of validation samples.

**Vector Scaling (VS)** (Guo et al., 2017) extends TS by introducing class-specific temperature scaling and bias parameters. The calibrated logits $z_i'$ are computed as:

$$z_i' = z_i \odot \mathbf{t} + \mathbf{b} \tag{14}$$

where: $\mathbf{t} \in \mathbb{R}^K$ is the vector of temperature scaling parameters for each class, $\mathbf{b} \in \mathbb{R}^K$ is the vector of bias parameters for each class, and $\odot$ denotes element-wise multiplication. Therefore, the parameters $\mathbf{t}$ and $\mathbf{b}$ are learned by minimizing the NLL loss on the validation set:

$$\min_{\mathbf{t},\mathbf{b}} \mathcal{L}_{\text{NLL}}(\mathbf{t}, \mathbf{b}) = -\frac{1}{N} \sum_{i=1}^{N} \log \left( \text{softmax} \left( z_i \odot \mathbf{t} + \mathbf{b} \right)_{y_i} \right) \tag{15}$$

ETS (Zhang et al., 2020) combines multiple calibration methods by weighting their outputs to improve calibration performance. Specifically, ETS combines the predictions from temperature-scaled logits, uncalibrated logits, and a uniform distribution. The calibrated probability vector $p_i$ for sample $i$ is computed as:

$$p_i = w_1 \cdot \text{softmax} \left( \frac{z_i}{T} \right) + w_2 \cdot \text{softmax}(z_i) + w_3 \cdot \mathbf{u} \tag{16}$$

where: $z_i \in \mathbb{R}^K$ is the vector of logits for sample $i$. $T > 0$ is the temperature parameter learned from TS, $w_1, w_2, w_3 \geq 0$ are the ensemble weights, satisfying $w_1 + w_2 + w_3 = 1$, and $\mathbf{u} = \frac{1}{K}\mathbf{1} \in \mathbb{R}^K$ is the uniform probability distribution over $K$ classes.

The ensemble weights $\mathbf{w} = [w_1, w_2, w_3]$ are optimized by minimizing the NLL loss:

$$\min_{\mathbf{w}} \quad \mathcal{L}_{\text{NLL}}(\mathbf{w}) = -\frac{1}{N} \sum_{i=1}^{N} \log \left( p_{i,y_i} \right)$$
$$\text{subject to} \quad \sum_{j=1}^{3} w_j = 1, \quad w_j \geq 0 \quad \text{for } j = 1, 2, 3 \tag{17}$$

**CaGCN** (Wang et al., 2021) incorporates graph structure into the calibration process by applying a GCN to the logits to compute node-specific temperatures. The temperature $T_i$ for node $i$ is computed as:

$$T_i = \text{softplus}\left(f_{\text{GCN}}(z, A)_i\right) \tag{18}$$

where $z \in \mathbb{R}^{N \times K}$ is the matrix of logits for all nodes, $A$ is the adjacency matrix of the graph, $f_{\text{GCN}}(\cdot)$ is the GCN function mapping from logits to temperature logits, $\text{softplus}(\cdot)$ ensures the temperatures are positive, and $T_i$ is scalar for node $i$. The calibrated logits follows 12: Note that CaGCN scales the logits multiplicatively with $T_i$, where $T_i > 0$, to ensure its accuracy preserving.

The GCN parameters $\theta$ are learned by minimizing the NLL loss on the validation set:

$$\min_\theta \ \mathcal{L}_{\text{NLL}}(\theta) = -\frac{1}{N} \sum_{i=1}^{N} \log\left(\text{softmax}\left(z_i'\right)_{y_i}\right) \tag{19}$$

**GATS** (Hsu et al., 2022) employs a GAT to compute node-specific temperatures, taking into account both the graph structure and node features. The notations are nearly the same as CaGCN, with a difference in the model architecture, from $f_{\text{GCN}}$ to $f_{\text{GAT}}$. The number of attention heads is set to 2.

### A.2 DATASET SOURCE DESCRIPTIONS

We evaluated our method on several widely used benchmark datasets, all accessible via the Deep Graph Library (DGL)[1]. These datasets encompass a variety of graph types and complexities, allowing us to assess the robustness and generalizability of our calibration approach.

**Citation Networks (Cora, Citeseer, Pubmed, Cora-Full)**: In these datasets (Sen et al., 2008; McCallum et al., 2000; Giles et al., 1998), nodes represent academic papers, and edges denote citation links between them. Node features are typically bag-of-words representations of the documents, capturing the presence of specific words. Labels correspond to the research topics or fields of the papers. The *Cora-Full* dataset is an extended version of *Cora*, featuring more nodes and a larger number of classes, which introduces additional classification challenges.

**Coauthor Networks (Coauthor CS, Coauthor Physics)**: These datasets (Shchur et al., 2018) represent co-authorship graphs where nodes are authors, and edges indicate collaboration between authors. Node features are derived from the authors' published papers, reflecting their research interests. Labels represent the most active field of study for each author. These datasets are larger and have higher average degrees compared to the citation networks, testing the models' ability to handle more densely connected graphs.

**Amazon Co-Purchase Networks (Computers, Photo)**: In these graphs (Yang & Leskovec, 2012), nodes represent products on Amazon, and edges connect products that are frequently co-purchased. Node features are extracted from product reviews and descriptions, providing rich textual information. Labels correspond to product categories. These datasets exhibit strong community structures and higher connectivity, offering a different perspective from academic networks.

**Ogbn-arxiv**: Part of the Open Graph Benchmark (Hu et al., 2020), this dataset is a large-scale citation network of ArXiv papers. Nodes represent papers, edges denote citation relationships, node features are obtained from paper abstracts, and labels are assigned based on subject areas. Its size and complexity make it suitable for evaluating scalability and performance on real-world, large graphs.

**Reddit**: This dataset (Hamilton et al., 2017) models interactions on Reddit. Nodes are posts, and edges connect posts if the same user comments on both, capturing user interaction patterns. Node features are derived from post content and metadata, and labels correspond to the community (subreddit) of each post. Its large size and dense connectivity challenge models to scale efficiently while maintaining performance.

| Dataset | Uncalibrated ($\times 10^{-3}$) | GETS ($\times 10^{-3}$) | Improvement (%) |
|---|---|---|---|
| **Citeseer** | $4.46 \pm 0.82$ | $2.95 \pm 1.08$ | 33.9% |
| **Computers** | $8.44 \pm 1.11$ | $5.86 \pm 3.02$ | 30.6% |
| **Cora-full** | $10.02 \pm 0.15$ | $4.87 \pm 0.62$ | 51.4% |
| **Cora** | $2.07 \pm 0.17$ | $1.69 \pm 0.28$ | 18.3% |
| **CS** | $2.05 \pm 0.13$ | $2.05 \pm 0.18$ | 0.0% |
| **Ogbn-arxiv** | $1.81 \pm 0.29$ | $4.72 \pm 0.65$ | - |
| **Photo** | $5.12 \pm 0.78$ | $2.62 \pm 0.30$ | 48.8% |
| **Physics** | $1.29 \pm 0.12$ | $1.46 \pm 0.20$ | - |
| **Pubmed** | $4.69 \pm 0.28$ | $1.63 \pm 0.23$ | 65.3% |
| **Reddit** | $26.38 \pm 0.45$ | $16.51 \pm 1.54$ | 37.4% |

Table 6: Comparison of VarECE between Uncalibrated and GETS models, with improvement in percentage. Positive values indicate improvement.

### A.3 ALGORITHMIC FAIRNESS AMONG DEGREE GROUPS

While fairness is not the primary focus of this paper, our model enhances algorithmic fairness by incorporating graph structural information into the calibration process, improving fairness across different degree groups. Although grouping nodes based on degrees is a nonparametric binning strategy—which may diverge from our main focus on parametric calibration—we include this discussion to address calibration errors among these groups. This demonstrates that our method promotes fairness by achieving similar calibration results across nodes with varying degrees. The metrics and analyses presented can be extended to algorithmic fairness studies to evaluate model trustworthiness.

We define a binning strategy $\mathcal{B}^* = \{\mathcal{D}_1, \mathcal{D}_2, \ldots, \mathcal{D}_B\}$ that groups nodes based on their degrees, each containing approximately $\frac{N}{B}$ nodes, where $N$ is the total number of nodes. The average degree of group $\mathcal{D}_i$ is: $\text{AvgDeg}(\mathcal{D}_i) = \frac{1}{|\mathcal{D}_i|} \sum_{v \in \mathcal{D}_i} \deg(v)$, where $\deg(v)$ is the degree of node $v$. Groups are ordered such that $\text{AvgDeg}(\mathcal{D}_i) < \text{AvgDeg}(\mathcal{D}_j)$ for $i < j$. Note that $\mathcal{B}^*$ can also be defined using other topological statistics.

Following the *Rawlsian Difference Principle* (Rawls, 1971; Kang et al., 2022), we aim to balance calibration errors across degree-based groups by minimizing their variance. We define the Variation of ECE (VarECE) as a fairness metric:

$$\text{VarECE}(f_\theta, \mathcal{B}^*) = \text{Var}\left( \left\{ \frac{|\mathcal{D}_j|}{N} |\text{Acc}(\mathcal{D}_j) - \text{Conf}(\mathcal{D}_j)| \right\}_{j=1}^{B} \right), \tag{20}$$

where $\text{Acc}(\mathcal{D}_j)$ and $\text{Conf}(\mathcal{D}_j)$ are the accuracy and confidence on group $\mathcal{D}_j$, respectively.

Results are given in Table 6. By minimizing VarECE, our model promotes fairness across different groups, ensuring that no group—particularly those with lower degrees—suffers from disproportionately higher calibration errors.

### A.4 ABLATION OF ENSEMBLE STRATEGIES

If we ablate the input ensemble, we still use $M = 7$, but only use the logit $z$ for the input. In this case, we have trained $M$ same calibration models and do the average.

Table 7: Ablation studies on different input configurations, measured by ECE.

| Input-Ablation | Citeseer | Computers | Cora | Cora-full | CS | Ogbn-arxiv | Photo | Physics | Pubmed | Reddit |
|---|---|---|---|---|---|---|---|---|---|---|
| GETS | $6.70 \pm 1.60$ | $4.36 \pm 1.79$ | $2.95 \pm 0.43$ | $3.42 \pm 0.53$ | $1.78 \pm 0.10$ | $2.36 \pm 0.11$ | $2.03 \pm 0.34$ | $1.08 \pm 0.09$ | $1.87 \pm 0.31$ | $2.86 \pm 0.49$ |

If we ablate the model ensemble, the diverse input type is concatenated and is fed into one GNN calibrator, which is reduced to the idea GATS, which is demonstrated to perform less efficiently and effectively than GETS in our previous discussion of Table 2.

---

[1]https://github.com/shchur/gnn-benchmark

Table 8: GPU reserved memory usage for each algorithm. The OOM case is marked with the attempted reserved memory from the algorithm.

| Dataset | Classifier | TS | VS | ETS | CaGCN | GATS | GETS |
|---|---|---|---|---|---|---|---|
| **Citeseer** | GCN | 164 MB | 164 MB | 118 MB | 164 MB | 168 MB | 184 MB |
| **Computers** | GCN | 210 MB | 210 MB | 190 MB | 212 MB | 348 MB | 250 MB |
| **Cora** | GCN | 98 MB | 98 MB | 98 MB | 98 MB | 102 MB | 110 MB |
| **Cora-full** | GCN | 1380 MB | 1380 MB | 1380 MB | 1380 MB | 1384 MB | 2018 MB |
| **CS** | GCN | 1546 MB | 1546 MB | 620 MB | 1060 MB | 1066 MB | 1546 MB |
| **Ogbn-arxiv** | GCN | 3382 MB | 3382 MB | 3382 MB | 1328 MB | 2858 MB | 2418 MB |
| **Photo** | GCN | 186 MB | 186 MB | 130 MB | 154 MB | 216 MB | 186 MB |
| **Physics** | GCN | 2368 MB | 2368 MB | 2368 MB | 2368 MB | 2390 MB | 3504 MB |
| **Pubmed** | GCN | 224 MB | 224 MB | 224 MB | 166 MB | 176 MB | 224 MB |
| **Reddit** | GCN | 7208 MB | 7208 MB | 7208 MB | 7206 MB | 17.54 GB | 7208 MB |

### A.5 CALIBRATION RELIABILITY DIAGRAM RESULTS

Based on the classifier GCN, the reliability diagrams reflect how the confidence aligns with the accuracies in each bin. The reliability diagram of calibration results of different models is given in Figure 4 and 5. The ECE calculated by degree-based binning is shown in Figure 6.

### A.6 GPU MEMORY USAGE

We summarize the GPU memory reserved by each of the calibration methods. By default, all the calibration methods run on the GCN classifier outputs. The memory usage results are shown in Table 8.

### A.7 SIMCALIB COMPARISON

We further compare with the work of SimCalib (Tang et al., 2024), as shown in Table 9.

Table 9: ECE Comparison of GETS and SimCalib results for various datasets and classifiers. Smaller values are highlighted in bold.

| Dataset | Classifier | GETS | SimCalib |
|---|---|---|---|
| Cora | GCN | **2.29 ± 0.52** | 3.32 ± 0.99 |
| | GAT | **2.00 ± 0.48** | 2.90 ± 0.87 |
| Citeseer | GCN | **2.50 ± 1.42** | 3.94 ± 1.12 |
| | GAT | **1.98 ± 0.30** | 3.95 ± 1.30 |
| Pubmed | GCN | 1.90 ± 0.40 | **0.93 ± 0.32** |
| | GAT | **0.78 ± 0.15** | 0.95 ± 0.35 |
| Computers | GCN | 2.03 ± 0.35 | **1.37 ± 0.33** |
| | GAT | 1.77 ± 0.27 | **1.08 ± 0.33** |
| Photo | GCN | 2.07 ± 0.31 | **1.36 ± 0.59** |
| | GAT | **1.10 ± 0.25** | 1.29 ± 0.55 |
| CS | GCN | 1.34 ± 0.10 | **0.81 ± 0.30** |
| | GAT | 1.05 ± 0.27 | **0.83 ± 0.32** |
| Physics | GCN | 0.87 ± 0.09 | **0.39 ± 0.14** |
| | GAT | **0.29 ± 0.05** | 0.40 ± 0.13 |
| CoraFull | GCN | 3.32 ± 1.24 | **3.22 ± 0.74** |
| | GAT | **1.52 ± 2.27** | 3.40 ± 0.91 |

### A.8 GATING SCORES VISUALIZATION

We also visualize the stack plot of the gating score changes during training. This will reflect how different experts are preferred during different stages of training. By default, we look into the GETS-GCN case run on the GCN classifier.

## A.9 ABLATION ON THE INPUT ENSEMBLE

### A.9.1 DIFFERENT STRUCTURAL INFORMATION

We conducted a test to replace degree embedding with betweenness centrality embedding, page rank centrality, clustering coefficient, harmonic centrality, Katz centrality, and Node2Vec embedding. We conduct experiments on all datasets, but some of the datasets are too large to compute the network statistics. We left those datasets empty due to limited time in rebuttal periods.

The results are shown in Table 10. For naming convenience, we use GETS-centrality and GETS-Node2Vec to represent different structural information. The original GETS is also repeated here for comparison. By default, we use the classifier GCN. For Node2Vec embedding, we set by default `walk_length`=20, `num_walks`=10, `workers`=4

Table 10: Ablation studies on different structural information, measured by ECE (cleaned values).

| Expert | Citeseer | Computers | Cora | Cora-full | CS | Ogbn-arxiv | Photo | Physics | Pubmed | Reddit |
|---|---|---|---|---|---|---|---|---|---|---|
| **GETS** | $2.50 \pm 1.42$ | $2.03 \pm 0.35$ | $2.29 \pm 0.52$ | $3.32 \pm 1.24$ | $1.34 \pm 0.10$ | $1.85 \pm 0.22$ | $2.07 \pm 0.31$ | $0.87 \pm 0.09$ | $1.90 \pm 0.40$ | $1.49 \pm 0.07$ |
| **GETS-Betweenness** | $7.63 \pm 1.35$ | $5.01 \pm 3.49$ | $3.29 \pm 0.75$ | $3.63 \pm 0.68$ | $2.21 \pm 0.90$ | / | $1.84 \pm 0.17$ | / | $2.18 \pm 0.35$ | / |
| **GETS-Clustering** | $7.56 \pm 0.88$ | $3.54 \pm 2.30$ | $3.48 \pm 0.41$ | $3.61 \pm 0.77$ | $1.80 \pm 0.08$ | / | $2.00 \pm 0.46$ | $1.15 \pm 0.16$ | $2.18 \pm 0.30$ | / |
| **GETS-PageRank** | $7.72 \pm 1.27$ | $3.88 \pm 1.38$ | $3.25 \pm 0.80$ | $3.55 \pm 0.95$ | $1.83 \pm 0.11$ | / | $1.73 \pm 0.12$ | $1.12 \pm 0.12$ | $2.43 \pm 0.37$ | / |
| **GETS-Node2Vec** | $4.25 \pm 1.28$ | $2.99 \pm 0.97$ | $3.06 \pm 0.54$ | $3.88 \pm 1.06$ | $1.82 \pm 0.09$ | / | $1.85 \pm 0.40$ | $1.13 \pm 0.15$ | $2.30 \pm 0.33$ | / |

### A.9.2 DIFFERENT INPUT TYPES

We further ablate on different input types. We ablate one of the three input types and create the input ensemble based on the other two. For example, we ablate $z$ and then construct the input ensemble as $\{\mathbf{x}, d, [\mathbf{x}, d]\}$. For the convenience of naming, we use GETS-DX, GETS-DZ, and GETS-XZ to represent the input ensemble without including $z$, $\mathbf{x}$, and $d$. The results can be found in Table 11. Generally, incorporating more information in the input types would make the results better.

Table 11: Ablation studies on different input ensembles, measured by ECE (cleaned values).

| Expert | Citeseer | Computers | Cora | Cora-full | CS | Ogbn-arxiv | Photo | Physics | Pubmed | Reddit |
|---|---|---|---|---|---|---|---|---|---|---|
| **GETS** | $2.50 \pm 1.42$ | $2.03 \pm 0.35$ | $2.29 \pm 0.52$ | $3.32 \pm 1.24$ | $1.34 \pm 0.10$ | $1.85 \pm 0.22$ | $2.07 \pm 0.31$ | $0.87 \pm 0.09$ | $1.90 \pm 0.40$ | $1.49 \pm 0.07$ |
| **GETS-DX** | $4.00 \pm 0.94$ | $2.76 \pm 0.40$ | $3.27 \pm 0.62$ | $3.60 \pm 0.54$ | $1.86 \pm 0.21$ | $2.15 \pm 0.46$ | $2.12 \pm 0.42$ | $1.06 \pm 0.08$ | $2.02 \pm 0.35$ | $1.55 \pm 0.25$ |
| **GETS-DZ** | $4.29 \pm 1.24$ | $3.53 \pm 2.36$ | $2.66 \pm 0.71$ | $3.34 \pm 0.34$ | $1.83 \pm 0.15$ | $2.32 \pm 0.27$ | $1.76 \pm 0.58$ | $0.98 \pm 0.12$ | $2.43 \pm 0.56$ | $1.80 \pm 0.16$ |
| **GETS-XZ** | $2.93 \pm 0.72$ | $3.49 \pm 1.47$ | $3.09 \pm 0.47$ | $3.26 \pm 0.40$ | $1.94 \pm 0.52$ | $2.24 \pm 0.19$ | $1.33 \pm 0.15$ | $1.02 \pm 0.07$ | $2.56 \pm 0.55$ | $2.28 \pm 0.84$ |

## A.10 ABLATION ON THE EXPERTS

We extend the Table 3 by including Multi-layer Perceptron (MLP) as the backbone model with the same layer number for training the algorithm. MLP is structure-unaware, which can serve as the baseline. The MLP results show that the selection of the backbone methods for the experts does not necessarily require graph-based models, which offer the opportunity to include a broader range of models for the graph calibration task.

Table 12: Ablation studies on different expert models, measured by ECE.

| Expert | Citeseer | Computers | Cora | Cora-full | CS | Ogbn-arxiv | Photo | Physics | Pubmed | Reddit |
|---|---|---|---|---|---|---|---|---|---|---|
| **GETS** | $2.50 \pm 1.42$ | $2.03 \pm 0.35$ | $2.29 \pm 0.52$ | $3.32 \pm 1.24$ | $1.34 \pm 0.10$ | $1.85 \pm 0.22$ | $2.07 \pm 0.31$ | $0.87 \pm 0.09$ | $1.90 \pm 0.40$ | $1.49 \pm 0.07$ |
| **GETS-GAT** | $4.09 \pm 0.71$ | $3.64 \pm 1.94$ | $2.96 \pm 0.70$ | $14.04 \pm 5.70$ | $4.91 \pm 3.93$ | $1.61 \pm 0.28$ | $3.43 \pm 1.82$ | $2.57 \pm 2.23$ | $1.96 \pm 0.59$ | OOM |
| **GETS-GIN** | $4.34 \pm 1.36$ | $4.56 \pm 3.33$ | $5.53 \pm 0.59$ | $2.83 \pm 0.46$ | $2.29 \pm 0.82$ | $2.48 \pm 0.30$ | $4.06 \pm 2.96$ | $1.16 \pm 0.14$ | $2.30 \pm 0.58$ | $4.64 \pm 1.03$ |
| **GETS-MLP** | $4.82 \pm 0.85$ | $4.09 \pm 1.51$ | $3.19 \pm 0.65$ | $3.51 \pm 1.02$ | $1.89 \pm 0.10$ | $2.38 \pm 0.17$ | $1.38 \pm 0.46$ | $0.99 \pm 0.15$ | $2.18 \pm 0.33$ | $1.96 \pm 0.48$ |

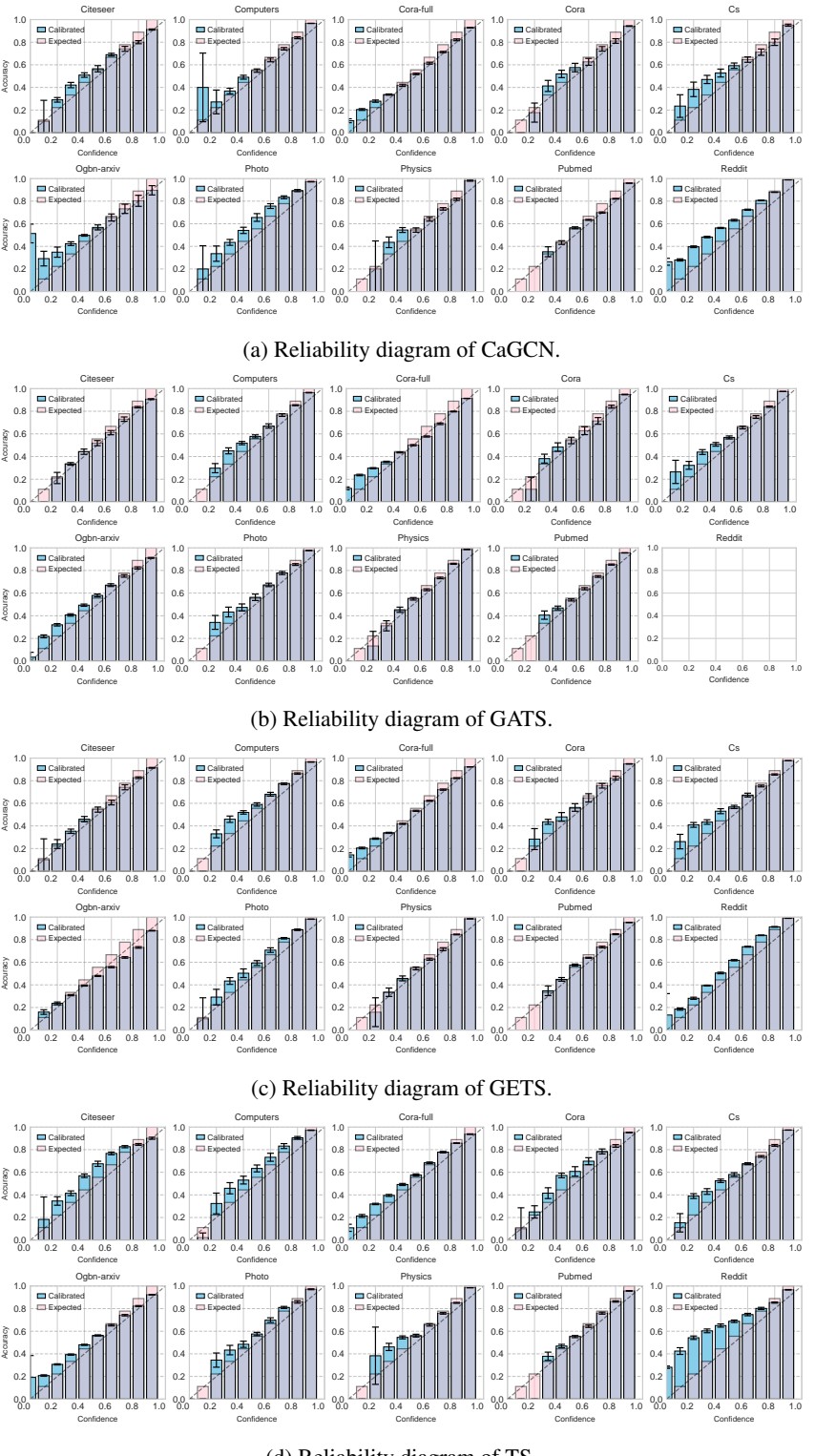

Figure 4: Reliability diagrams of GaGCN, GATS, GETS, and TS.

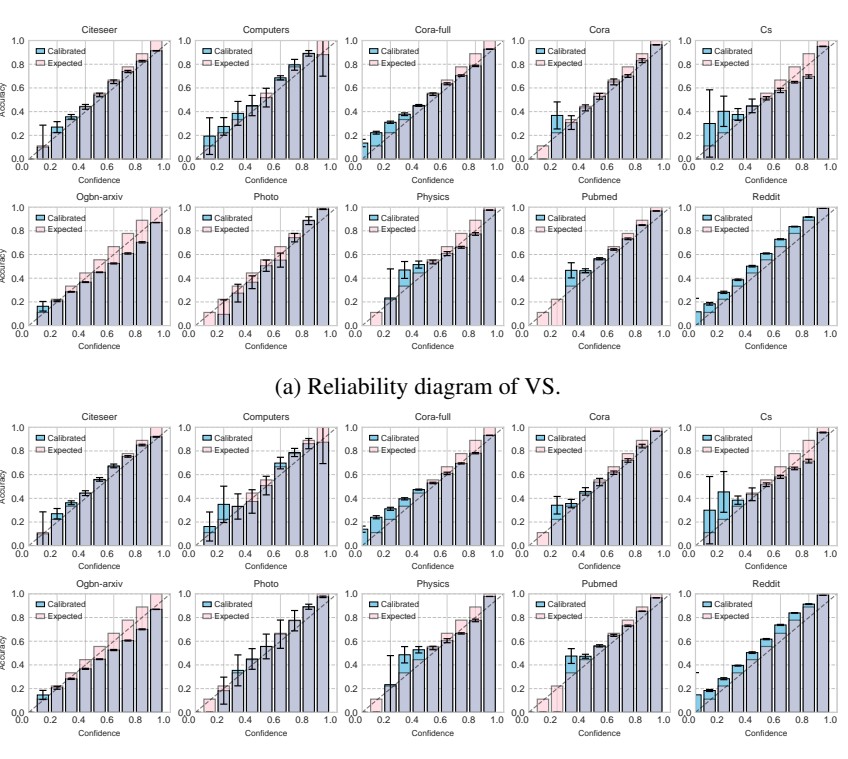

(a) Reliability diagram of VS.

(b) Reliability diagram of ETS.

Figure 5: Reliability diagrams of VS and ETS.

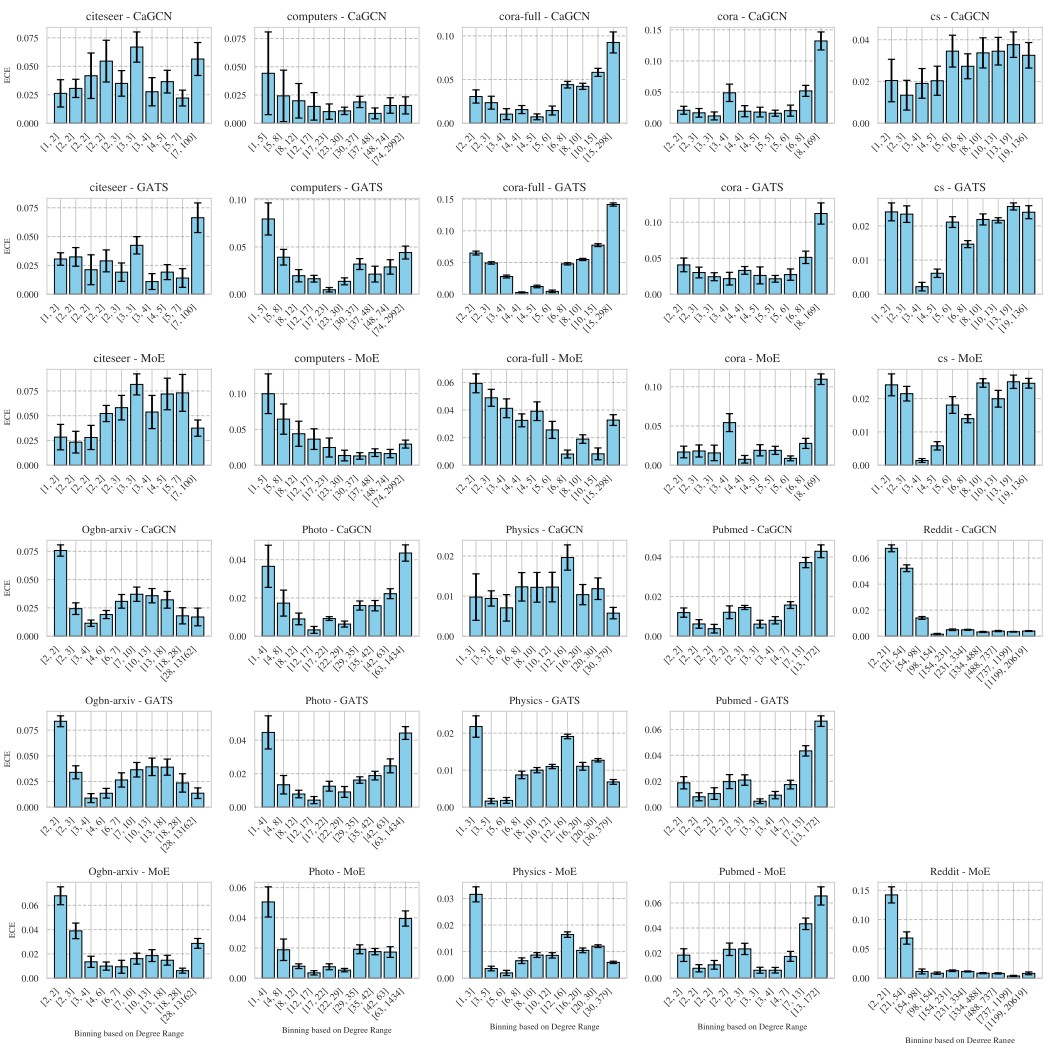

Figure 6: Visualization of the ECE performance of three deep learning models (CaGCN, GATS, and GETS) across various datasets. Confidence bins are sorted by degree. Note that GATS encountered an out-of-memory issue on the Reddit dataset, resulting in a blank figure for that case.

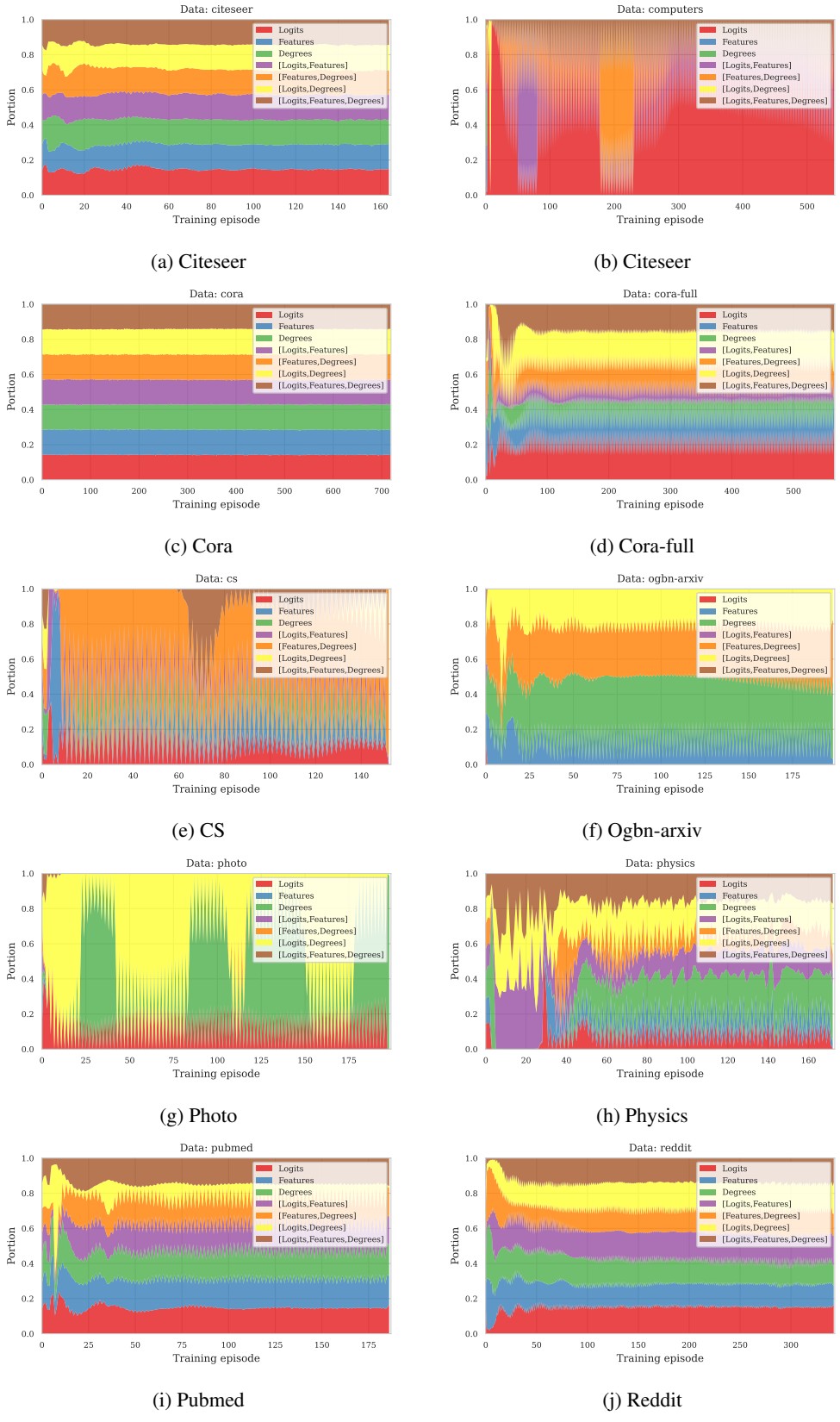

Figure 7: GETS gating scores changes during training.

