# OpenReview forum: "GETS: Ensemble Temperature Scaling for Calibration in Graph Neural Networks"
_ICLR.cc/2025/Conference — ICLR 2025 Spotlight_

### Official Review · Reviewer_aQtF · 2024-11-01

**Soundness:** 3
**Presentation:** 3
**Contribution:** 3
**Rating:** 8
**Confidence:** 4

**Summary:**

This paper introduces Graph Ensemble Temperature Scaling (GETS), a calibration framework for Graph Neural Network that combines input and model ensemble strategies with a Graph Mixture-of-Experts architecture. GETS integrates multiple types of inputs and adaptively selects relevant experts for calibrating such node's predictions. The framework achieves improvements in calibration performance across 10 benchmark datasets.

**Strengths:**

* The comprehensive experimental results in Table 2 show consistent improvements across different architectures and datasets.
* The sparse activation mechanism described in Section 4.2 enables efficient scaling to larger graphs.
* The well-presented analysis of expert selection patterns provides insights into which input combinations are most effective for calibration for each dataset.

**Weaknesses:**

* The hyperparameter sensitivity analysis is somewhat limited, say regarding the number of experts, or the expert embedding dimension. For example,
 (1) How do calibration performance and computation costs scale with the number of experts (M) ?
 (2) What is the impact of expert network capacity (e.g. the number of neurons, the number of layers) on calibration quality?
 (3) How does the choice of top-k in sparse gating affect the trade-off between performance and efficiency? An empirical experiment on synthetic dataset would provide valuable implementation guide.

* While computational efficiency is emphasize, the need to store multiple expert models could be problematic for resource-constrained settings. This practical limitation deserves more discussion. For instance, computational memory requirements for different types of experts handling various input might be helpful.

**Questions:**

1. Could the authors provide quantitative analysis of expert specialization during the training? This could be potentially done by
 (1) Visualization of expert activation patterns (investigation of gating function values) across training epochs, or
 (2) Per-expert contribution to the total calibration loss over training epochs, or
 (3) Show how the preference for certain experts emerges and stabilizes (again through analyzing the gating function G or gating scores Q)
While Figure 3(b) demonstrates the final expert selection patterns through sparse activation, a dynamic analysis of expert contribution to performance would be valuable.

2. How does the architecture of individual experts impact calibration performance? Given that experts process different input types, different architecture may be more suitable for each input modality. Currently, GCN with 2 layers is used for all types except the integer degree. Does the same GCN architecture uniformly reflect the effects across the experts of different types?

3. How does GETS perform on heterogeneous graphs where nodes have different feature types or varying degrees of connectivity?

---

> ### Author Response · Authors · 2024-11-23
> **Reply to Questions 1,2,3**
>
> Thank you very much for sharing insightful viewpoints.
> 1. For the first question, we are doing further visualization and attaching the picture and comments in the revised manuscript. It will be revised sooner; once it is finished, we will add more remarks in the comment box.
> 2. Thank you for the valuable comments. We admit that there is still more room to enhance our single GCN structure, e.g., GIN and GCNII. We did not implement this since when comparing the GETS with other SOTA in the GNN calibration field, the prior arts mainly used 2-layer GCN (one algorithm uses GAT). To ensure the fairness of the baseline comparison, we adopt GCN in this paper experiment. We will further explore this field in future work, especially investigating how the graph network structure (i.e., attention, transformer) influences the calibration output performance.
> 3. Thank you for your deep thought on extending our methodology to the heterogeneous graphs. Similar to doing graph learning on heterogeneous graphs, we can do calibration tasks based on the selected input via the classical methodology described in https://dl.acm.org/doi/10.1145/3292500.3330961. We will also extend our framework to this type of graph and see the performance inherited from its specific structure.

---

> > ### Author Response · Authors · 2024-11-23
> > **New Experiment Results for Quesion 1**
> >
> > Thank you for pointing out the interesting direction in question 1. Following your guidance, we did some visualization tailored to the points you mentioned. You can check the updated manuscript: Manuscript-> Appendix-> Section A.8 (Page 17). We will update you with more results once our experiment is completed. Thanks for your attention.

---

> ### Author Response · Authors · 2024-11-25
>
> We thank you again for the time and effort you dedicated to reviewing our work. We wanted to kindly follow up to inquire if you have had the opportunity to review our response.
>
> If there are any remaining concerns or questions, we would be happy to discuss them further and do our best to address them. If our responses have satisfactorily addressed your concerns, we would greatly appreciate your reconsideration of the score.
>
> Thanks,
>
> The Authors

---

> > ### Comment · Reviewer_aQtF · 2024-11-29
> >
> > Dear Authors,
> >
> > Thanks for addressing my questions. The visualization looks great. I think it would be more insightful if you could connect some of the graph statistics with their convergence pattern. For example, computers, CS, and Photo dataset, their focus on specific inputs vary a lot during the training procedure while others seem to stabilize.

---

> > > ### Author Response · Authors · 2024-11-29
> > > **Correlating Gating Score Variability with Training Loss Convergence**
> > >
> > > Dear Reviewer aQtF,
> > >
> > > Thank you very much for your valuable comments and insightful feedback! Based on your suggestions, we have expanded our analysis to include convergence patterns by incorporating the training loss over each epoch and comparing it with the gating scores.
> > >
> > > To address your observation regarding variability, we focused on the differences in gate score dynamics for datasets like computers, CS, and Photo, which exhibit significant variability, versus more stable datasets like Cora. Specifically, we analyzed the correlation between the standard deviation of gating scores across different input types and the training loss.
> > >
> > > We have updated our results and uploaded the notebook scripts used for generating these figures to our anonymous GitHub repository. You can review them at the following link:
> > >
> > > https://anonymous.4open.science/r/GETS/rebuttal/gating_score_vs_training_loss/README.md
> > >
> > > Thank you again for your constructive feedback, which has significantly improved the quality of our work.

---

> > > > ### Author Response · Authors · 2024-11-29
> > > > **Follow-up of Gating Score Variability with Training Loss Convergence**
> > > >
> > > > Dear Reviewer aQtF,
> > > >
> > > > The key findings of the experiments include:
> > > > * Datasets like `computers`, `CS`, and `photo` show significant variability in node gate focus during the training process, aligning with rapid changes in training loss.
> > > > * Datasets like `pubmed` and `physics` exhibit early stabilization, with minimal variability in node gates and smoother training loss trends.
> > > >
> > > > These findings provide a deeper understanding of how variability in node gates corresponds to different convergence behaviors across datasets. Thank you again for your constructive feedback!

---

### Official Review · Reviewer_uyzv · 2024-11-03

**Soundness:** 3
**Presentation:** 4
**Contribution:** 2
**Rating:** 6
**Confidence:** 3

**Summary:**

This work leverages multiple GNNs with different inputs within a Graph Mixture-ofExperts (MoE) architecture for confidence calibration of GNNs.

**Strengths:**

- Paper is written well and easy to read.

- The proposed framework is intuitional and the design choices are justified.

- Experimental results are thorough and enough empirical analyses are provided.

- Proposed method seems to perform well compared to the selected baselines.

**Weaknesses:**

- More recent baselines can be included to further strengthen experimental results.

- The novelties of the proposed framework seem to be rather limited based on the current presentation. It would be better to clearly emphasize the novelties over the existing solutions in the Related Work.

**Questions:**

Please see weaknesses.

---

> ### Author Response · Authors · 2024-11-23
> **Reply to Weakness 1,2**
>
> 1. Many thanks for pointing out the problems. We are currently doing more baseline experiment comparisons. We will update the results in the manuscript soon. Once it is finished, I will add some remarks in the comment box.
> 2. Thanks for pointing out this problem. We apologize for not clarifying our novelty. Our novelty lies in threefold:
> (i)Empirical Finding: We find the imbalance calibration results over nodes with different degrees, which was not pointed out in the previous literature.
> (ii)A Comprehensive Rethinking and Redesign of GNN Calibration Task: We rethink the calibration algorithm motivated by our theoretical findings. Previous calibration algorithms on GNN either use a new graph network without carefully designing the input feature or focus on limited influential factors for the calibration problem. We naturally follow the idea of calibrating GNN by a new GNN, then diving into how to choose input features (we summarize three diverse input spaces by literature review and our experiment) and how to aggregate/handle multiple features inputs (we innovatively introduce the graph MOE to this problem for performance enhancement and generalizability). A comprehensive benchmark is provided accordingly. For this question, we will follow your suggestion and add more precise literature review in the revised manusciprt.

---

> ### Author Response · Authors · 2024-11-23
> **More Experiment Result in A.7**
>
> Thank you for pointing out the lack of several works in our baseline comparison. In this rebuttal phase, we did a comprehensive review of the literature again and incorporated another very recent framework (https://arxiv.org/abs/2312.11858) into our baseline comparison. You can check the updated manuscript: Manuscript-> Appendix-> Section A.7 (Page 17). We will update you with more results once our experiment is completed. Thanks for your attention.

---

> ### Author Response · Authors · 2024-11-25
>
> We thank you again for the time and effort you dedicated to reviewing our work. We wanted to kindly follow up to inquire if you have had the opportunity to review our response.
>
> If there are any remaining concerns or questions, we would be happy to discuss them further and do our best to address them. If our responses have satisfactorily addressed your concerns, we would greatly appreciate your reconsideration of the score.
>
> Thanks,
>
> The Authors

---

> ### Author Response · Authors · 2024-11-30
>
> We thank you again for the time and effort you dedicated to reviewing our work. We are following up to inquire if you have had the opportunity to review our response. Since the rebuttal deadline is coming, we appreciate your valuable feedback.
>
> We would be happy to discuss any remaining concerns or questions further and do our best to address them. If our responses have satisfactorily addressed your concerns, we would greatly appreciate your reconsidering the score.
>
> Thanks,
>
> The Authors

---

> ### Author Response · Authors · 2024-12-02
>
> We thank you again for the time and effort you dedicated to reviewing our work. We are following up to inquire if you have had the opportunity to review our response. Since the rebuttal deadline is coming, we appreciate your valuable feedback.
>
> We would be happy to discuss any remaining concerns or questions further and do our best to address them. If our responses have satisfactorily addressed your concerns, we would greatly appreciate your reconsidering the score.
>
> Thanks,
>
> The Authors

---

### Official Review · Reviewer_XSU8 · 2024-11-03

**Soundness:** 2
**Presentation:** 3
**Contribution:** 3
**Rating:** 8
**Confidence:** 4

**Summary:**

This paper addresses a critical challenge in graph node classification: the issue of calibration. The authors propose a  method that integrates a combination of features, logits, and node degree within a framework of expert models, aiming to enhance the calibration performance in graph model. By innovatively focusing on the calibration aspect, the study offers new insights into improving the reliable of graph neural networks.

**Strengths:**

The authors propose a method that leverages node features, logits, and node degree, combined with a framework of  expert models, to enhance the reliability of node predictions. Through extensive experiments, the authors demonstrate the effectiveness of their method compared to existing baselines.

**Weaknesses:**

1. This paper lacks adequate theoretical backing for why using features, logits, and degree with experts can effectively address calibration issues. Additionally, it does not explain why other node factors are not considered. This needs to be clarified.

2. The paper does not explain how the chosen experts can resolve calibration issues, nor does it discuss why a separate model is not designed for each input. This lack of innovation and theoretical justification may undermine the reliability of the research.

3. The baseline models used in the experiments, particularly the graph calibration model baselines, are outdated. For instance, CAGCN was published in NeurIPS 2021 and GATS in 2022. It is recommended that the authors update their baseline models to enhance the comparability and validity of their approach.

**Questions:**

My questions can be found in weakness.

---

> ### Author Response · Authors · 2024-11-23
> **Reply to Weaknesses 1,2,3**
>
> Thanks for pointing out this meaningful direction.
> 1. For the question on the theoretical foundation of input feature selection,  we want to explain the problems from two perspectives. Firstly, it is still unknown what the key(essential) influential factors are that influence the calibration coefficient results in the GNN calibration network. Unlike traditional learning tasks (e.g., classification, prediction, regression) on the graph with raw input features, there is no direct mapping from features to logit smoothing parameters. On the contrary, analyzing influential factors of calibration tasks theoretically can be challenging: graph structure, logits before the activation layer, and other graph features are correlated with each other and are not independent. In the literature, various works have tried to analyze this problem empirically or theoretically, e.g., We chose three factors in our work since they represent three ‘diverse’ input spaces: (i). The feature vector represents the raw graph feature data, which was proved to be effective for calibration. In the previous work, low-dimension input features can greatly guide the calibration tree algorithm. (ii). The degree embedding represents the raw graph geometry and structure, which coincides with our observation of the imbalance phenomena. Intuitively, in the learning task, the decision based on little message passing could have a high probability of outputting unreliable results (although it can be right in accuracy level) (iii). The logits represent the model after training (which is different from raw data and structure), which was proved to be efficient empirically by observing the total variation gap. We admit that there is still room to explore the possible combination of several input factors, and this will be our future direction. In the mixture-of-expert system, we need to control the expert number reasonably since if we give a lot of experts and just let the gating select the appropriate one, we can imagine that the generalization may be better, but the limited training samples can also lead to a poor understanding of each expert. We will continuously investigate the problem you
> mentioned before in future works.
> 2. The expert in our framework is used to (i). Represent the calibrator for node-wise calibration (ii). Act as a choice in the MOE system to enhance the calibration performance and generalizability across various graph data.
> Reason: By explaining our intuition and answers to question 1, it can be inferred that the best selection of feature input is still not perfectly solved. In addition, in the graph structure, different nodes' confidence may possess different dependencies on features. In this way, it is not wise to aggregate features from different spaces mentioned in 1 into a unified embedding space. Motivated by the success of modern LLM, we propose a mixture of expert-based calibration. The high-level idea is: For a specific node,  let a new gating network choose a calibrator and the corresponding feature for calibration on this node. It enhances the calibration performance and its generalization on different graph datasets (since the mechanism of choosing experts can adapted to different graphs). In addition, the structure of MOE is direct, and more experts could be added once we find a more competitive expert.
> 3. We have added more results in our revised manuscript. You can check it out in the appendix sooner. We will also update the table in the new reply! Many thanks!

---

> ### Author Response · Authors · 2024-11-23
> **More Experiment in Appendix A.7**
>
> Thank you for pointing out the lack of several works in our baseline comparison. In this rebuttal phase, we did a comprehensive review of the literature again and incorporated another very recent framework (https://arxiv.org/abs/2312.11858) into our baseline comparison. You can check the updated manuscript: Manuscript-> Appendix-> Section A.7 (Page 17). We will further update more results once our experiment is completed, thanks for your attention.

---

> ### Author Response · Authors · 2024-11-25
>
> We thank you again for the time and effort you dedicated to reviewing our work. We wanted to kindly follow up to inquire if you have had the opportunity to review our response.
>
> If there are any remaining concerns or questions, we would be happy to discuss them further and do our best to address them. If our responses have satisfactorily addressed your concerns, we would greatly appreciate your reconsideration of the score.
>
> Thanks,
>
> The Authors

---

> ### Author Response · Authors · 2024-11-30
>
> We thank you again for the time and effort you dedicated to reviewing our work. We are following up to inquire if you have had the opportunity to review our response. Since the rebuttal deadline is coming, we appreciate your valuable feedback.
>
> We would be happy to discuss any remaining concerns or questions further and do our best to address them. If our responses have satisfactorily addressed your concerns, we would greatly appreciate your reconsidering the score.
>
> Thanks,
>
> The Authors

---

> ### Comment · Reviewer_XSU8 · 2024-12-02
>
> I agree to raise my score, since authors have addressed my concerns.

---

> > ### Author Response · Authors · 2024-12-02
> > **Appreciation for Reviewer Insights**
> >
> > Dear Reviewer XSU8,
> >
> > We sincerely thank the reviewers for their thoughtful and constructive feedback, which has greatly contributed to improving the quality and clarity of our work. We deeply appreciate the time and effort you dedicated to providing such detailed and insightful comments, and we are grateful for your willingness to reconsider our revised manuscript. Your valuable input has been instrumental in strengthening our research.
> >
> > Thanks,
> > Authors

---

### Official Review · Reviewer_sXvB · 2024-11-03

**Soundness:** 3
**Presentation:** 3
**Contribution:** 3
**Rating:** 8
**Confidence:** 4

**Summary:**

The paper proposes graph ensemble temperature scaling (GETS), a novel calibration technique for node-classification GNNs.
GETS is a parametric calibration method which calibrates the predictions for each node in a graph by predicting a temperature for each node which is used as a factor that is applied to the respective class logits.
GETS predicts node-wise temperatures using a set of GNN-based calibration models trained on different features (predicted logits, node degrees, node features or combinations thereof). The temperatures returned by the GNN calibrators are then combined via a learned gating function which selects a top-k subset of the calibrators for each vertex.

To summarize, GETS combines the previously proposed ensemble temperature scaling (ETS) method with the idea of using a GNN to predict structure-aware temperatures (e.g., CaGCN or GATS).

**Strengths:**

As reported in the experiments, the combination of structural information and a mixture of calibrators trained on different sets of features seems to clearly outperform previously proposed calibration techniques for node classifiers.

This combination of ideas is well motivated by the paper and the claims regarding the advantages of GETS are supported by the broad range of used datasets.

Presentation-wise, the paper is well structured and the ideas are presented concisely and clearly.

**Weaknesses:**

As mentioned, the reported advantages of the GETS approach seem to stem from the combination of two ideas:
1. Predicting node-wise temperatures based on class logits, node features and node degrees.
2. Incorporating structural dependencies between vertices to predict temperatures.

Both ideas have (to some extent) been utilized independently in prior work. While the performed experiments show that the combination of both ideas works well in practice, it only gives limited insight into why exactly this is the case.

More specifically, I believe it would have been interesting to investigate the following questions:
1. *What is the influence of the three types of features used by the experts ($z$, $x$ and $d$)?*
   Figure 3b only partially answers this question by showing how frequently the different experts are selected by the gating function. It does not show to which extent the different features are able to substitute each other, i.e., it would be interesting to investigate ablations of GETS without different subsets of experts/features being part of the pool.
2. *How well does a GETS-like MoE approach with a structure-unaware MLP instead of a GNN work?* While the evaluated ETS calibration approach also uses the idea of combining multiple calibrators, those calibrators do not use the combinations of $z$, $x$ and $d$ used by GETS. Isolating the effect of this choice of features on the calibration performance would be interesting. A related work in this direction by Liu et al. (2022) [1] could be considered here.
3. *Are other structural features than the degree count $d$ also useful for calibration?* There is an overlap in the information provided by the node degree inputs and the graph convolutions performed by each expert. As an addition to the previous point, it would be interesting to investigate whether other structural features (e.g., other node centrality measures or node2vec-like embeddings) can serve as effective surrogates to the structural information that is incorporated via the graph convolutions in each calibrator.

By addressing (at least some) of these questions, I believe, that the paper could shed a more nuanced light on the importance of different features for calibration in the node classification setting.

---
[1] Liu, T. et al.: On Calibration of Graph Neural Networks for Node Classification. In: 2022 International Joint Conference on Neural Networks (IJCNN). pp. 1–8 (2022). [https://doi.org/10.1109/IJCNN55064.2022.9892866](https://doi.org/10.1109/IJCNN55064.2022.9892866).

**Questions:**

Apart from the suggestions/questions described in the weaknesses, I only have two minor questions:
1. You report that the GATS calibration approach suffers from OOM problems not present in the other methods. In Section 5.3 you note that the main difference between CaGCN and GATS, time-complexity-wise, lies in the addition of the attention heads $H$. If all computations are parallelized in memory, I would therefore assume the space complexity to be similar. However, in supplement A1.3, you state that you used $H=2$, which appears to be reasonably small, especially considering the fact that $H$ is only a linear factor in the complexity. Does GATS really require orders of magnitude more memory than the other approaches or were the OOM-issues caused by GATS requiring slightly more memory than available? If the former, do you have an explanation as to why this is the case?
2. As I understand, GETS performs temperature scaling (TS) for each vertex. In your experiments, you also evaluated vector scaling (VS). Have you also considered a variant of GETS that uses VS instead of TS?

---

> ### Author Response · Authors · 2024-11-23
> **Reply to the weakness and questions**
>
> Many thanks for pointing out our weaknesses and sharing your thoughts for further improvement.
> Weakness:
> 1. For the first and the third weakness of feature selection,  we want to explain the problems from two perspectives. Firstly, it is still unknown what the key(essential) influential factors are that influence the calibration coefficient results in the GNN calibration network. Unlike traditional learning tasks (e.g., classification, prediction, regression) on the graph with raw input features, there is no direct mapping from features to logit smoothing parameters. On the contrary, analyzing influential factors of calibration tasks theoretically can be challenging: graph structure, logits before the activation layer, and other graph features are correlated with each other and are not independent. In the literature, various works have tried to analyze this problem empirically or theoretically, e.g., We chose three factors in our work since they represent three ‘diverse’ input spaces: (i). The feature vector represents the raw graph feature data, which was proved to be effective for calibration. In the literature, low-dimension input features can greatly guide the calibration tree algorithm. (ii). The degree embedding represents the raw graph geometry and structure, which coincides with our observation of the imbalance phenomena. Intuitively, in the learning task, the decision based on little message passing could have a high probability of outputting unreliable results (although it can be right in accuracy level) (iii). The logits represent the model after training (which is different from raw data and structure), which was proved to be efficient empirically by observing the total gap. We admit that there is still room to explore the possible combination of several input factors, and this will be our future direction. In the mixture-of-expert system, we need to control the expert number in a reasonable way since if we give a lot of experts and just let the gating select the appropriate one, we can imagine that the generalization may be better, but the limited training samples can also lead to a poor understanding of each expert. We will continuously investigate the problem you mentioned before in future works.
> 2. Regarding the second weakness of the usage of GNN in calibration tasks, we appreciate this discussion since it is highly related to the key idea of the paper. We use GNN for calibration, which is motivated by the graph messaging mechanism and connectivity structure. The tricky problem lies in the fact that we need to explore the correct feature input and how to use it reasonably, which is the core of this paper. For graph datasets, we are used to handling problems via graph neural nets; it will be interesting to investigate the MLP version without structure.
>
> Questions:
> 1. For the first question reporting complexity issues. We believe that the computational complexity grows linearly according to attention head. One technique point we are not sure about is how to do memory parallel in the GATS setting (We did not do so in our implementation). It will be challenging and interesting to investigate the reason for the OOM problem. We made several attempts and showed some results using the CUDA information. (We will update this sooner in the reply as well as the updated manuscript)
> 2. For the question on applying MOE to VS. Yes, we can apply our novel MOE ensemble techniques to the vector scaling mechanism. The variable to do MOE becomes a matrix and a vector, not the smoothing parameter t in the TS setting.  However, the reason we do not consider this variant in the experiment is that it does not hold accuracy-preserving properties, which is in contrast with our initial demand in the calibration task.

---

> ### Author Response · Authors · 2024-11-23
> **Evidence of OOM problem**
>
> Dear reviewer, thank you for pointing out the OOM problem and leading us to investigate it further. Theoretically, the complexity is linear dependent on the attention head. Under the H=2 setting, we try to understand this OOM by documenting the reserved storage for Cuda. (This has been inserted in the manuscript, as Table 8 shows). We find that in the OOM dataset, the GPU memory allocation will explode in the reserved memory (Case: Reddit-GATS-17.54GB).
>
> Based on the experiment results, we have two insights/explanations for the OOM problem. 1. The graph edges highly determine the memory issues in the GNN calibrator. In the Reddit dataset, the edge number is 11606919, which is a huge number. 2. For GATS, although its memory complexity is the attention head number $\times$ complexity of GCN, the constant multiplication can lead to OOM in the application (which is very different from theoretical analysis, where we omit the constant and denote them all as $\mathcal{O}(\cdot)$).
>
> We will continue to follow this line. Welcome any suggestions and comments.

---

> ### Author Response · Authors · 2024-11-25
>
> We thank you again for the time and effort you dedicated to reviewing our work. We wanted to kindly follow up to inquire if you have had the opportunity to review our response.
>
> If there are any remaining concerns or questions, we would be happy to discuss them further and do our best to address them. If our responses have satisfactorily addressed your concerns, we would greatly appreciate your reconsideration of the score.
>
> Thanks,
>
> The Authors

---

> > ### Comment · Reviewer_sXvB · 2024-11-26
> >
> > Thank you for your detailed response!
> >
> > **Re W1, W2, W3:** I agree that a deeper analysis of the influence of the different design choices made is interesting and should be considered in future work. Even though a broad-scale evaluation of the design space of graph calibration methods would be out of scope for your work, as mentioned in the review, I still believe that even a small-scale ablation study on some of the described aspects would strengthen your contribution.
> >
> > **Re Q1:** Thank you for investigating the OOM issue. If I understood correctly, the memory problem of GATS is in principle solvable by scaling up the memory linearly with the number of heads. Given a fixed number of heads, the memory requirements of GATS and GETS are equivalent, up to some constant scaling factor. In Section 5.3 (line 415), you state that "GETS proves to be more scalable than GATS"; I find this statement to be slightly misleading since the memory growth behavior of both approaches is, in fact, identical. Thus, GETS is not necessarily more "scalable" than GATS (depending on the definition of that word) but simply more memory efficient.
> >
> > **Re Q2:** Thank you! This fully answers my question. I did not consider that the accuracy-preserving property was one of your stated goals.

---

> ### Author Response · Authors · 2024-11-27
> **Thank you for the valuable comments**
>
> Thank you for the valuable comments. We did a series of ablation studies according to your advice, and the following are the results. (towards your follow-up W1,W2,W3). **We also modify our manuscript accordingly. You can check the experiment setup and naming description in the current manuscript, Appendix A.9. (Since the answer box has maximum character limits, we can not explain in this box, sorry for this)** There are still some experiments running; once we finish the task, we will post the results.
>
> 1. Ablation studies on different structural information measured by ECE (cleaned values).
>
> | **Expert**               | **Citeseer**      | **Computers**     | **Cora**          | **Cora-full**     | **CS**           | **Ogbn-arxiv** | **Photo**        | **Physics**      | **Pubmed**       | **Reddit**       |
> |-------------------------|-------------------|-------------------|-------------------|-------------------|-------------------|-----------------|------------------|------------------|------------------|------------------|
> | **GETS**                | 2.50 ± 1.42       | 2.03 ± 0.35       | 2.29 ± 0.52       | 3.32 ± 1.24       | 1.34 ± 0.10       | 1.85 ± 0.22     | 2.07 ± 0.31      | 0.87 ± 0.09      | 1.90 ± 0.40      | 1.49 ± 0.07      |
> | **GETS-Centrality**     | 7.63 ± 1.35       | 5.01 ± 3.49       | 3.29 ± 0.75       | 3.63 ± 0.68       | 2.21 ± 0.90       | /               | 1.84 ± 0.17      | /                | 2.18 ± 0.35      | /                |
> | **GETS-Node2Vec**       | 4.25 ± 1.28       | 2.99 ± 0.97       | 3.06 ± 0.54       | 3.88 ± 1.06       | 1.82 ± 0.09       | /               | 1.85 ± 0.40      | 1.13 ± 0.15      | 2.30 ± 0.33      | /                |
> 2. Ablation studies on different input ensembles, measured by ECE (cleaned values)
>
> | **Expert**               | **Citeseer**      | **Computers**     | **Cora**          | **Cora-full**     | **CS**           | **Ogbn-arxiv** | **Photo**        | **Physics**      | **Pubmed**       | **Reddit**       |
> |-------------------------|-------------------|-------------------|-------------------|-------------------|-------------------|-----------------|------------------|------------------|------------------|------------------|
> | **GETS**                | 2.50 ± 1.42       | 2.03 ± 0.35       | 2.29 ± 0.52       | 3.32 ± 1.24       | 1.34 ± 0.10       | 1.85 ± 0.22     | 2.07 ± 0.31      | 0.87 ± 0.09      | 1.90 ± 0.40      | 1.49 ± 0.07      |
> | **GETS-DX**             | 4.00 ± 0.94       | 2.76 ± 0.40       | 3.27 ± 0.62       | 3.60 ± 0.54       | 1.86 ± 0.21       | 2.15 ± 0.46     | 2.12 ± 0.42      | 1.06 ± 0.08      | 2.02 ± 0.35      | 1.55 ± 0.25      |
> | **GETS-DZ**             | 4.29 ± 1.24       | 3.53 ± 2.36       | 2.66 ± 0.71       | 3.34 ± 0.34       | 1.83 ± 0.15       | 2.32 ± 0.27     | 1.76 ± 0.58      | 0.98 ± 0.12      | 2.43 ± 0.56      | 1.80 ± 0.16      |
> | **GETS-XZ**             | 2.93 ± 0.72       | 3.49 ± 1.47       | 3.09 ± 0.47       | 3.26 ± 0.40       | 1.94 ± 0.52       | 2.24 ± 0.19     | 1.33 ± 0.15      | 1.02 ± 0.07      | 2.56 ± 0.55      | 2.28 ± 0.84      |
> 3. Ablation studies on different expert models measured by ECE.
>
> | **Expert**               | **Citeseer**      | **Computers**     | **Cora**          | **Cora-full**     | **CS**           | **Ogbn-arxiv** | **Photo**        | **Physics**      | **Pubmed**       | **Reddit**       |
> |-------------------------|-------------------|-------------------|-------------------|-------------------|-------------------|-----------------|------------------|------------------|------------------|------------------|
> | **GETS**                | 2.50 ± 1.42       | 2.03 ± 0.35       | 2.29 ± 0.52       | 3.32 ± 1.24       | 1.34 ± 0.10       | 1.85 ± 0.22     | 2.07 ± 0.31      | 0.87 ± 0.09      | 1.90 ± 0.40      | 1.49 ± 0.07      |
> | **GETS-GAT**            | 4.09 ± 0.71       | 3.64 ± 1.94       | 2.96 ± 0.70       | 14.04 ± 5.70      | 4.91 ± 3.93       | 1.61 ± 0.28     | 3.43 ± 1.82      | 2.57 ± 2.23      | 1.96 ± 0.59      | OOM              |
> | **GETS-GIN**            | 4.34 ± 1.36       | 4.56 ± 3.33       | 5.53 ± 0.59       | 2.83 ± 0.46       | 2.29 ± 0.82       | 2.48 ± 0.30     | 4.06 ± 2.96      | 1.16 ± 0.14      | 2.30 ± 0.58      | 4.64 ± 1.03      |
> | **GETS-MLP**            | 4.82 ± 0.85       | 4.09 ± 1.51       | 3.19 ± 0.65       | 3.51 ± 1.02       | 1.89 ± 0.10       | 2.38 ± 0.17     | 1.38 ± 0.46      | 0.99 ± 0.15      | 2.18 ± 0.33      | 1.96 ± 0.48      |

---

> > ### Author Response · Authors · 2024-11-27
> > **Ablation setup and naming description**
> >
> > Table 1:
> >
> > We conduct the test to replace the degree embedding with centrality embedding and Node2Vec embedding. We conduct experiments on all datasets, but some of the datasets are too large to compute the network statistics. We left those datasets empty due to limited time in rebuttal periods. For the naming convenience, we use GETS-centrality and GETS-Node2Vec to represent different structural information. The original GETS is also repeated here for comparison. By default, we use the classifier GCN. For Node2Vec embedding, we set by default walk_length=20, num_walks=10, workers=4
> >
> > Table 2:
> >
> > We further ablate on different input types. We ablate one of the three input types and create the input ensemble based on the other two. For example, we ablate $z$ and then construct the input ensemble as $\{\mathbf{x},d,[\mathbf{x},d]\}$. For the convenience of naming, we use GETS-DX, GETS-DZ, and GETS-XZ to represent the input ensemble without including $z$, $\mathbf{x}$, and $d$. Generally, incorporating more information in the input types would make the results better.
> >
> > Table 3:
> >
> > We include Multi-layer Perceptron (MLP) as the backbone model with the same layer number for training the algorithm. MLP is structure-unaware, which can serve as the baseline. The MLP results show that the selection of the backbone methods for the experts does not necessarily require graph-based models, which offer the opportunity to include a broader range of models for the graph calibration task.

---

> > > ### Comment · Reviewer_sXvB · 2024-11-27
> > >
> > > Thank you for conducting those additional experiments so quickly! Overall they indeed seem to support the claims made in the paper and show that the used combination of different structural features is helpful for the considered task.
> > >
> > > As you point out, your results also show that the use of a GNN is not always necessary to achieve good performance, indicating that fixed structural input features can be sufficient. Nonetheless, the use of a GNN still seems to be beneficial most of the time.
> > >
> > > I have one follow-up question regarding Table 1. You wrote that you use a "centrality embedding" instead of a "degree embedding" for the "GETS-Centrality" ablation. I find this to be confusing, since degree centrality is also one of many centrality measures for graphs - which other centrality measure(s) did you use for the "GETS-Centrality" model?

---

> > > > ### Author Response · Authors · 2024-11-27
> > > > **Thank you for the valuable comments**
> > > >
> > > > Thank you for your valuable comments and suggestions. We want to make some explanations, and we are very sorry for the confusion we caused.
> > > >
> > > > For your question on centrality embedding, in our current version, we use degree embedding for the experiment setup. In addition, we are currently running other centrality measures using our framework. One interesting finding (also the reason we have not included the results in the rebuttal system) is that the algorithm takes much more time to train when using other centrality embeddings than the simplest degree embedding. (It requires several hours on our server system for large graphs.)
> > > >
> > > > We will report the experiment to you once they are done. We will list the centrality name and the corresponding results. From the empirical computational cost viewpoint, we can also infer that **degree** is a lightweight centrality measure for our graph calibration task, achieving both good performance and efficient training time.
> > > >
> > > > Thank you again for your comments.

---

> > > > > ### Author Response · Authors · 2024-11-28
> > > > > **Further explanations and experiments on the centrality embedding**
> > > > >
> > > > > Thanks for your valuable comments. As mentioned, we added the centrality embedding: clusteing coefficient and pagerank. The result is shown below. We hope to receive your further comments and reviews. Many thanks
> > > > >
> > > > > | Expert | Citeseer | Computers | Cora | Cora-full | CS | Ogbn-arxiv | Photo | Physics | Pubmed | Reddit |
> > > > > |--------|----------|-----------|------|-----------|----|-----------:|-------|---------|--------|--------|
> > > > > | GETS | 2.50 ± 1.42 | 2.03 ± 0.35 | 2.29 ± 0.52 | 3.32 ± 1.24 | 1.34 ± 0.10 | 1.85 ± 0.22 | 2.07 ± 0.31 | 0.87 ± 0.09 | 1.90 ± 0.40 | 1.49 ± 0.07 |
> > > > > | GETS-Degree | 7.63 ± 1.35 | 5.01 ± 3.49 | 3.29 ± 0.75 | 3.63 ± 0.68 | 2.21 ± 0.90 | / | 1.84 ± 0.17 | / | 2.18 ± 0.35 | / |
> > > > > | GETS-Clustering | 7.56 ± 0.88 | 3.54 ± 2.30 | 3.48 ± 0.41 | 3.61 ± 0.77 | 1.80 ± 0.08 | / | 2.00 ± 0.46 | 1.15 ± 0.16 | 2.18 ± 0.30 | / |
> > > > > | GETS-PageRank | 7.72 ± 1.27 | 3.88 ± 1.38 | 3.25 ± 0.80 | 3.55 ± 0.95 | 1.83 ± 0.11 | / | 1.73 ± 0.12 | 1.12 ± 0.12 | 2.43 ± 0.37 | / |
> > > > > | GETS-Node2Vec | 4.25 ± 1.28 | 2.99 ± 0.97 | 3.06 ± 0.54 | 3.88 ± 1.06 | 1.82 ± 0.09 | / | 1.85 ± 0.40 | 1.13 ± 0.15 | 2.30 ± 0.33 | / |

---

> > > > > > ### Comment · Reviewer_sXvB · 2024-11-28
> > > > > >
> > > > > > Thank you for your continued effort! I have one remaining question: How exactly does the GETS-Degree variant differ from the standard GETS approach? As I understood, GETS also uses the degree information, so I assume that you did not only swap out the degree features but also made other changes. Could you provide more details on those experiments?

---

> > > > > > > ### Author Response · Authors · 2024-11-28
> > > > > > > **Explain the difference between GETS and GETS-Degree**
> > > > > > >
> > > > > > > Thank you for your valuable feedback. We apologize for the confusion caused by our explanation! To clarify, during the implementation of GETS, the degree information is used by counting node degrees and feeding these counts into an *nn.Embedding* layer to achieve higher-dimensional representations.
> > > > > > >
> > > > > > > In contrast, GETS-Degree, which is inappropriately named, specifically utilizes **betweenness centrality**, which is computed using the *nx.betweenness_centrality()* function, and similarly undergoes nn.embedding. This approach is analogous to other centrality measures such as clustering coefficients, PageRank, and Node2Vec—each replaces the degree centrality embedding with an embedding of the respective centrality feature.
> > > > > > >
> > > > > > > To avoid further confusion, we rename GETS-Degree to GETS-Betweenness to more explicitly reflect that it leverages betweenness centrality. The updated table is listed below. We hope this clarification addresses your concerns and makes our methodology more transparent.
> > > > > > >
> > > > > > > | Expert | Citeseer | Computers | Cora | Cora-full | CS | Ogbn-arxiv | Photo | Physics | Pubmed | Reddit |
> > > > > > > |--------|----------|-----------|------|-----------|----|-----------:|-------|---------|--------|--------|
> > > > > > > | GETS | 2.50 ± 1.42 | 2.03 ± 0.35 | 2.29 ± 0.52 | 3.32 ± 1.24 | 1.34 ± 0.10 | 1.85 ± 0.22 | 2.07 ± 0.31 | 0.87 ± 0.09 | 1.90 ± 0.40 | 1.49 ± 0.07 |
> > > > > > > | GETS-Betweenness | 7.63 ± 1.35 | 5.01 ± 3.49 | 3.29 ± 0.75 | 3.63 ± 0.68 | 2.21 ± 0.90 | / | 1.84 ± 0.17 | / | 2.18 ± 0.35 | / |
> > > > > > > | GETS-Clustering | 7.56 ± 0.88 | 3.54 ± 2.30 | 3.48 ± 0.41 | 3.61 ± 0.77 | 1.80 ± 0.08 | / | 2.00 ± 0.46 | 1.15 ± 0.16 | 2.18 ± 0.30 | / |
> > > > > > > | GETS-PageRank | 7.72 ± 1.27 | 3.88 ± 1.38 | 3.25 ± 0.80 | 3.55 ± 0.95 | 1.83 ± 0.11 | / | 1.73 ± 0.12 | 1.12 ± 0.12 | 2.43 ± 0.37 | / |
> > > > > > > | GETS-Node2Vec | 4.25 ± 1.28 | 2.99 ± 0.97 | 3.06 ± 0.54 | 3.88 ± 1.06 | 1.82 ± 0.09 | / | 1.85 ± 0.40 | 1.13 ± 0.15 | 2.30 ± 0.33 | / |

---

> > > > > > > > ### Comment · Reviewer_sXvB · 2024-11-28
> > > > > > > >
> > > > > > > > Thank you! This clears things up. I have no further questions.

---

> > > > > > > > > ### Author Response · Authors · 2024-11-28
> > > > > > > > > **Clarifications Provided to Address Feedback and Enhance Evaluation**
> > > > > > > > >
> > > > > > > > > Thank you for your insightful feedback and for highlighting areas that could benefit from further clarification. We have addressed the misunderstanding regarding the implementation details and provided additional clarity on the model’s contributions through ablation studies. We believe these revisions effectively resolve the confusion and enhance the overall clarity of our methodology.
> > > > > > > > >
> > > > > > > > > We kindly hope that, with these clarifications, you might reconsider your evaluation. We deeply value the time and effort you have invested in reviewing our work and providing thoughtful comments. Thank you once again for your valuable input!

---

> > > > > > > > > > ### Comment · Reviewer_sXvB · 2024-11-28
> > > > > > > > > > **Final Remarks and Change of the Overall Rating**
> > > > > > > > > >
> > > > > > > > > > In my review I described three weaknesses of the original manuscript:
> > > > > > > > > >
> > > > > > > > > > 1. The influence of the different types of features used by the experts was not investigated.
> > > > > > > > > > 2. The relevance of the GNN module employed by GETS was unclear.
> > > > > > > > > > 3. The significance of using node degree as an additional structural feature was not made clear.
> > > > > > > > > >
> > > > > > > > > > In their rebuttal, the authors addressed all three weaknesses and performed a series of ablation studies, which show that the made design choices all contribute towards the observed performance.
> > > > > > > > > >
> > > > > > > > > > Since those additional results notably improve the contribution, I change my overall rating to accept.
> > > > > > > > > >
> > > > > > > > > > I thank the authors once again for the fruitful discussion!

---

> > > > > > > > > > > ### Author Response · Authors · 2024-11-28
> > > > > > > > > > > **Appreciation for Updated Remarks**
> > > > > > > > > > >
> > > > > > > > > > > Dear Reviewer sXvB28,
> > > > > > > > > > >
> > > > > > > > > > > Thank you for taking the time to reconsider our manuscript and for your valuable feedback throughout the review process. We appreciate your efforts in helping us improve our work.
> > > > > > > > > > >
> > > > > > > > > > > Best regards,
> > > > > > > > > > > The Authors

---

### Comment · Area_Chair_owYh · 2024-11-26

Dear reviewer aQtF, uyzv and XSU8,

Please ensure to inform the authors that you have checked their rebuttal. And if you have any additional concerns or questions based on their responses, please share them in time.

Thanks!

---

### Meta-Review · Area_Chair_owYh · 2024-12-18

**Metareview:**

The paper presents a novel approach, Graph Ensemble Temperature Scaling (GETS), aimed at improving the calibration of Graph Neural Networks (GNNs). By leveraging various input sources and employing a Mixture-of-Experts architecture, GETS effectively reduces expected calibration error across multiple benchmark datasets. This is a significant contribution, as it not only addresses a critical gap in existing calibration methods but also highlights the importance of accurate uncertainty estimation in high-stakes applications.

Reviewers presented mixed opinions regarding the paper's contributions and execution. While some reviewers appreciated the empirical results and potential improvements in calibration performance, others expressed concerns about the theoretical justification and the choice of baseline models used for comparison. Nonetheless, these differences did not impede the overall assessment of the paper's quality.

As the rebuttal phase progressed, the authors addressed the reviewers' concerns satisfactorily, clarifying the innovation in their approach and enhancing their experimental results. All reviewers ultimately expressed their satisfaction with the improvements made, leading to a consensus on the paper's merit. In conclusion, the reviewers unanimously agree that the quality of the paper is high and recommend acceptance based on its significant contributions and the authors' responsiveness to feedback.

**Additional Comments On Reviewer Discussion:**

Reviewers presented mixed opinions regarding the paper's contributions and execution. While some reviewers appreciated the empirical results and potential improvements in calibration performance, others expressed concerns about the theoretical justification and the choice of baseline models used for comparison. Nonetheless, these differences did not impede the overall assessment of the paper's quality.

As the rebuttal phase progressed, the authors addressed the reviewers' concerns satisfactorily, clarifying the innovation in their approach and enhancing their experimental results. All reviewers ultimately expressed their satisfaction with the improvements made, leading to a consensus on the paper's merit. In conclusion, the reviewers unanimously agree that the quality of the paper is high and recommend acceptance based on its significant contributions and the authors' responsiveness to feedback.

---

### Decision · Program_Chairs · 2025-01-22

Accept (Spotlight)